# Transcriptional profiling of matched patient biopsies clarifies molecular determinants of enzalutamide-induced lineage plasticity

Thomas C. Westbrook[1,18], Xiangnan Guan[2,18], Eva Rodansky[1], Diana Flores[1], Chia Jen Liu[3], Aaron M. Udager[3], Radhika A. Patel[4], Michael C. Haffner[4], Ya-Mei Hu[2], Duanchen Sun[2], Tomasz M. Beer[2], Adam Foye[5,6], Rahul Aggarwal[5,6], David A. Quigley[5,6], Jack F. Youngren[5,6], Charles J. Ryan[7], Martin Gleave[8], Yuzhuo Wang[8,9], Jiaoti Huang[10], Ilsa Coleman[4], Colm Morrissey[11], Peter S. Nelson[4], Christopher P. Evans[12], Primo Lara[12], Robert E. Reiter[13], Owen Witte[14], Matthew Rettig[13,15], Christopher K. Wong[16], Alana S. Weinstein[16], Vlado Uzunangelov[16], Josh M. Stuart[16], George V. Thomas[2], Felix Y. Feng[5,17], Eric J. Small[5,6], Joel A. Yates[1], Zheng Xia[2,19]✉ & Joshi J. Alumkal[1,19]✉

The androgen receptor (AR) signaling inhibitor enzalutamide (enza) is one of the principal treatments for metastatic castration-resistant prostate cancer (CRPC). Several emergent enza clinical resistance mechanisms have been described, including lineage plasticity in which the tumors manifest reduced dependency on the AR. To improve our understanding of enza resistance, herein we analyze the transcriptomes of matched biopsies from men with metastatic CRPC obtained prior to treatment and at progression ($n = 21$). RNA-sequencing analysis demonstrates that enza does not induce marked, sustained changes in the tumor transcriptome in most patients. However, three patients' progression biopsies show evidence of lineage plasticity. The transcription factor E2F1 and pathways linked to tumor stemness are highly activated in baseline biopsies from patients whose tumors undergo lineage plasticity. We find a gene signature enriched in these baseline biopsies that is strongly associated with poor survival in independent patient cohorts and with risk of castration-induced lineage plasticity in patient-derived xenograft models, suggesting that tumors harboring this gene expression program may be at particular risk for resistance mediated by lineage plasticity and poor outcomes.

Androgen deprivation therapy (ADT) is the principal treatment for metastatic prostate cancer, but progression to castration-resistant prostate cancer (CRPC) is nearly universal. In recent years, potent inhibitors of the androgen receptor (AR)—a luminal lineage transcription factor—have been developed, including the AR antagonist enzalutamide (enza)[1–5]. Enza improves progression-free survival and overall survival in patients with CRPC; further, enza also increases overall survival in patients with hormone-naive prostate cancer who are beginning ADT for the first time[6–8]. However, one-third of patients do not respond, and those with de novo resistance

have a significantly increased risk of death compared to responders[6–8].

Despite intense study, clinical enza resistance remains poorly understood. Several studies examined mechanisms of de novo or acquired enza resistance in clinical samples and implicated: *AR* amplification[9,10], *AR* splice variants[11,12], increased Wnt/β-catenin signaling[13–15], increased TGF-β signaling[14,16], epithelial to mesenchymal transition or increased stemness[14,17], and lineage plasticity[14]. However, these prior studies were largely restricted to DNA mutational profiling, compared baseline and progression samples from different patients, used limited numbers of matched samples, or did not focus on transcriptional changes.

Prior work suggests that most CRPC tumors resistant to AR signaling inhibitors (ARSIs) continue to depend on the AR[17,18]. However, we now appreciate that lineage plasticity—most commonly exemplified by loss of AR signaling and a switch from a luminal to an alternate differentiation program—is a resistance mechanism that appears to be increasing in the era of more widespread use of ARSIs[19]. The emergence of tumors with features of lineage plasticity may occur through diverse mechanisms: selection of a pre-existing clone that has already undergone differentiation change, acquisition of new genetic alterations that promote differentiation change, or transdifferentiation of tumor cells through epigenetic mechanisms[17,20–22].

Lineage plasticity is a continuum, ranging from tumors with persistent AR expression but low AR activity, those that lose AR expression but do not undergo neuroendocrine differentiation (double negative prostate cancer [DNPC]), and those that lose AR expression and do undergo neuroendocrine differentiation (neuroendocrine prostate cancer [NEPC])[23]. Importantly, CRPC tumors that have undergone lineage plasticity are associated with a much shorter survival than CRPC tumors that have persistent AR activity and a luminal lineage program, demonstrating an urgent need to understand treatment-induced lineage plasticity in prostate cancer[24].

In this work, we hypothesized that comparing gene expression profiles between matched CRPC tumor biopsy samples prior to enza and at the time of progression would identify pre-treatment and treatment-induced resistance mechanisms in individual patients. We describe results from 21 matched samples. We find evidence of lineage plasticity occurring in three of 21 progression tumors and define pathways and transcription factors that are highly activated in the baseline samples from patients whose tumors undergo lineage plasticity after enzalutamide treatment. Finally, we identify a gene signature associated with risk of therapy-induced lineage plasticity and poor patient survival.

## Results

### Heterogenous effects of enzalutamide treatment on the tumor transcriptome across matched biopsy samples

By examining the Stand Up to Cancer Foundation/Prostate Cancer Foundation West Coast Dream Team (WCDT) prospective cohort, we identified 21 patients with CRPC who underwent a metastatic tumor biopsy prior to enza and a repeat biopsy at the time of progression and whose tumor cells underwent RNA-sequencing after laser capture microdissection. All progression biopsies were performed prior to discontinuing enza, enabling us to identify resistance mechanisms induced by continued enza treatment.

The study design is shown in Fig. 1a. Patient demographic information and prior treatments are shown in Supplementary Table 1. Bone was the most common site for both pre-treatment and progression biopsies. Eighteen of 21 patients had the same tissue type biopsied at progression. In eight patients, the exact same lesion was biopsied at baseline and progression (Fig. 1b, Supplementary Table 2). The median time on enza treatment was 226 days, shorter than previous trials conducted in this same disease state[6,25]. PSA response at

12 weeks and the time between biopsies for each patient are shown in Fig. 1c.

To understand sample-to-sample differences, we performed unsupervised hierarchical clustering and found the nearest neighbor of 13/21 (62%) baseline samples was their matched progression sample pair (Fig. 2a). Samples did not cluster together based solely on the site of biopsy, indicating laser capture microdissection removed much of the microenvironment from these samples. Furthermore, whether the same lesion was biopsied did not impact how samples clustered.

We next examined measurements of interest in all the matched samples (Fig. 2b). To estimate AR transcriptional activity, we used Virtual Inference of Protein-activity by Enriched Regulon (VIPER) master regulator analysis[26]. Nine (43%) patients did not have a marked difference in inferred AR activity. Nine (43%) patients had decreased AR activity, and three (14%) patients had increased AR activity at progression (Supplementary Fig. 1a). We used a second method to measure AR activity—the ARG10 signature[27]. ARG10 strongly correlated with the VIPER results (Supplementary Fig. 1b). Though *AR-V7* expression increased in several samples at progression, the difference in expression using the entire 21-patient cohort was not statistically significant (Supplementary Fig. 1c).

Previously, Aggarwal et al. identified five clusters of CRPC tumors by RNA-sequencing analysis[24]. Cluster 2 was enriched for tumors with loss of AR activity and increased E2F1 activity and contained a preponderance of tumors that had lost AR expression[24], consistent with lineage plasticity. A subset of cluster 2 tumor samples was labeled NEPC based upon their morphologic appearance resembling small cell prostate cancer, though many of these tumor samples did not express canonical NEPC markers such as chromogranin A (CHGA) or synaptophysin (SYP)[24].

In examining the RNA-sequencing results from the baseline tumors, four of the five Aggarwal clusters were represented (clusters 1, 3, 4, and 5) in at least one sample, while no baseline sample harbored a cluster 2 program. We also applied the Labrecque transcription-based classifier that was developed on rapid autopsy CRPC samples and identified five subsets of prostate cancer: AR-driven prostate cancer (ARPC), amphicrine prostate cancer with neuroendocrine gene expression concomitant with AR signaling, AR-activity low prostate cancer, DNPC, and NEPC[23]. The Labrecque classifier designated all the baseline samples in our cohort as ARPC.

To determine if any of the progression tumors in our cohort underwent lineage plasticity after enza, we determined the Aggarwal cluster and Labrecque classifier designation. Twelve of 21 matched pairs did not change their Aggarwal cluster designation. However, three of the 21 progression tumors (hereafter referred to as converters) had gene expression profiles consistent with cluster 2, suggesting enza-induced conversion to an alternate lineage. We also examined the Labrecque classifier on the progression samples. The three converter samples designated as Aggarwal cluster 2 at progression were most consistent with DNPC by the Labrecque classifier, corroborating lineage plasticity in these tumors (Fig. 2b).

We next examined additional gene signatures linked previously to lineage plasticity in progression vs. baseline biopsies. Comparing samples from the three converter patients, signature scores for genes upregulated in NEPC tumors described by Beltran et al.[21] were increased (Supplementary Fig. 1d). A previously described basal stemness signature[28] was also activated in these three progression samples (Supplementary Fig. 1e). We previously identified a 76 gene AR-repressed gene signature that was activated in a CRPC cell line that undergoes enza-induced lineage plasticity[29]. This 76 gene signature was also increased in the progression samples from the three converters (Supplementary Fig. 1f). Finally, predicted AR activity was significantly decreased in the progression samples from the converters by

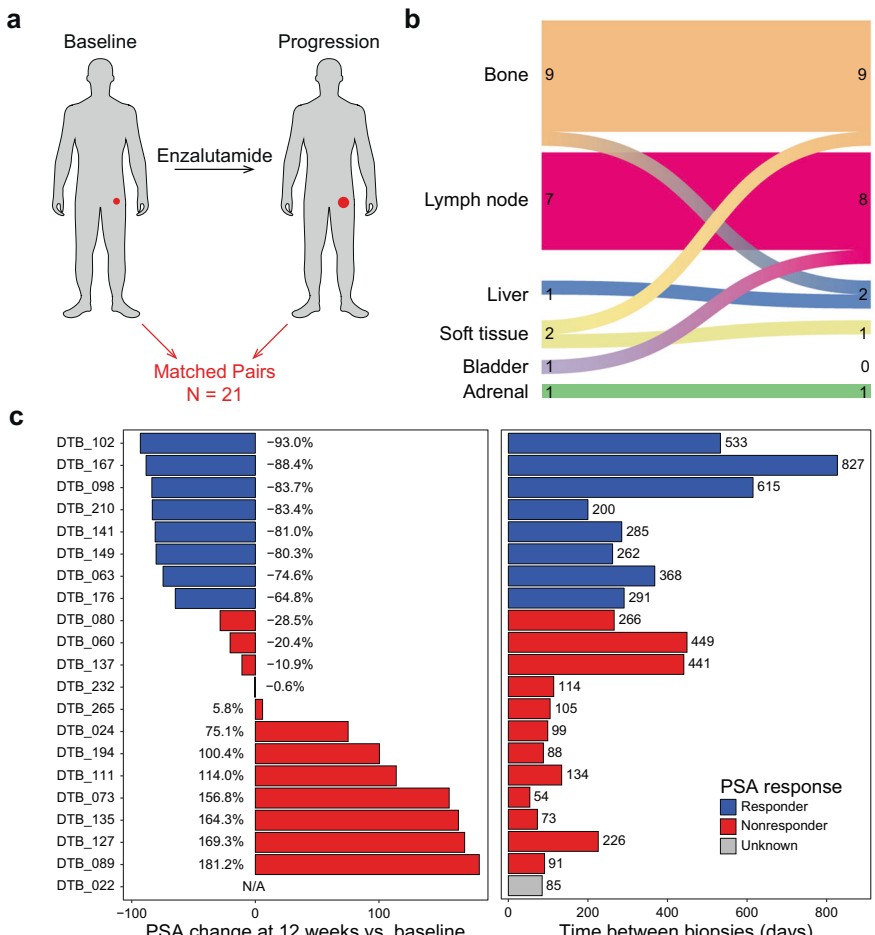

**Fig. 1 | Study biopsy and clinical information. a** Study schematic. **b** Sankey diagram showing site of biopsy at baseline (left) and at progression (right). Values indicate number of biopsies performed on each type of tissue at baseline (left) or progression (right). **c** Left panel shows PSA change at 12 weeks for each patient. Right panel shows time between biopsies for each patient. Response indicates whether a patient experienced a ≥ 50% reduction in PSA level at 12 weeks vs. baseline. For subject 022, PSA information was not available. Source data are provided as a Source Data file.

both VIPER and ARG10 signatures (Supplementary Fig. 1a, g). In examining pre- and post-treatment samples using the entire 21-patient cohort, none of these signatures was significantly changed, demonstrating that activation of these lineage plasticity signatures was not a generalized effect of enza treatment. Altogether, these results suggest that enza-induced lineage plasticity and conversion to an AR-independent program occurs in a subset of tumors (3/21 or 14%), similar to the frequency of cluster 2 tumors (10%) described by Aggarwal previously[24].

Notably, the baseline tumors from the three converter patients did not fall into the same Aggarwal cluster (cluster 4 for sample 80 and cluster 5 for samples 135, 210). Therefore, it was not surprising that the baseline tumors from these three patients did not cluster together using unsupervised clustering (Supplementary Fig. 1h, i). These data suggest that there may be different starting points to lineage plasticity with enza treatment.

### Clarification of a baseline transcriptional program linked to lineage plasticity risk

To identify genes linked with risk of lineage plasticity after enza, we examined the differentially expressed genes between the three baseline samples from converters vs. the 18 non-converters. Pathway analysis implicated activation of MYC targets, E2F targets, and allograft rejection in baseline tumors from converters (Fig. 3a). There were no significantly downregulated pathways in baseline tumors from converters. To identify differentially activated

transcription factors, we performed master regulator analysis. E2F1 was the top transcription factor predicted to be activated in the baseline tumors from converters (Fig. 3b, full list Supplementary Data 1), corroborating pathway analysis and our prior work demonstrating that high E2F1 activity is linked to lineage plasticity risk[29]. Additionally, we found that there was an upward trend in a previously described *RB1* loss signature[30] in the progression samples from converters, further suggesting E2F1 activation contributes to the lineage switch (Supplementary Fig. 1j). Other highly activated transcription factors in the baseline samples from converters include MYC family members and E2F4. Conversely, TP53—whose loss has been linked to lineage plasticity[27,31,32]—was predicted to be the most deactivated transcription factor (Fig. 3b).

Next, we focused on identifying genes significantly upregulated in the baseline tumors from converters vs. non-converters. We identified a 14-gene signature highly activated in the three baseline tumors from converters (Supplementary Table 3). Genes in this signature include those linked to: the Wnt pathway [RNF43[33] and TRABD2A[34]], the spliceosome [SNRPF[35]], and the electron transport chain [NDUFA12[36] and ATP5B[37]]. This signature trended downwards in the progression vs. baseline biopsies from the three converters (Supplementary Fig. 2a). These results suggest that this signature is not simply identifying tumor cells that have already undergone lineage plasticity prior to enza treatment. Rather, these genes may be markers of a transition state in cells susceptible to transcriptional conversion and lineage plasticity.

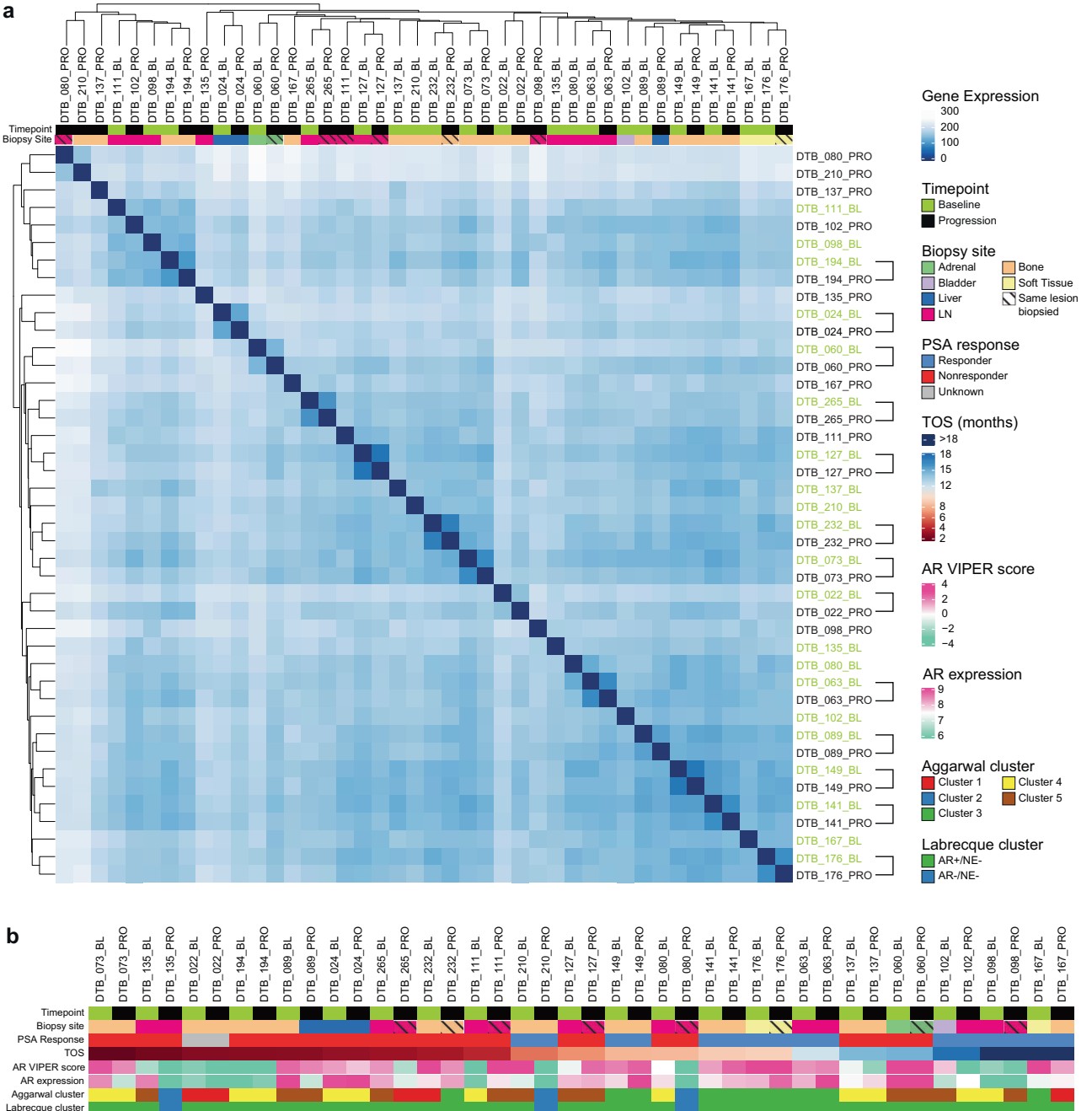

**Fig. 2 | The effect of enzalutamide on tumor transcriptome is heterogenous across patients. a** Similarity heatmap for all samples clustered by variance-stabilization transformation (vst). Hashes through biopsy site indicate that the same lesion was biopsied at baseline and progression. Bracket on right axis indicates that baseline and progression samples from the same patient are nearest neighbors. **b** Clinical and gene expression data for each matched pair ordered on x-axis by time between biopsies. TOS is time on study in months. AR expression is by log2(TPM + 1). Clusters for each sample were assigned based on classifications from Aggarwal et al.[24] and Labrecque et al.[23]. AR VIPER Score is the predicted AR activity score based on the AR regulon in the VIPER package[26]. Source data are provided as a Source Data file.

Dividing the baseline samples between converters and non-converters, we defined a cut off for this 14-gene lineage plasticity risk signature that separated the groups (Fig. 3c). Additional cohorts with matched biopsies before and after enza with lineage plasticity information are lacking. However, we hypothesized that patients whose baseline tumors had high scores for this lineage plasticity risk signature would have worse outcomes. Survival data from the time of ARSI treatment were available for several CRPC cohorts whose tumors had undergone RNA-sequencing—the International Dream Team dataset[9] and a prior prospective enza clinical trial led by our group[17]. Because a subset of the patients in that latter enza clinical

trial overlapped with the patients in this current report, we focused only on patients from that clinical trial not represented in this matched biopsy cohort. Using our pre-defined 14-gene signature score cut-off from the matched biopsy cohort, we determined that high scores were associated with worse overall survival from the time of ARSI treatment in both independent datasets ($p = 0.076$, $p = 0.005$; Fig. 3d, e). Thus, high expression of the 14-gene lineage plasticity risk signature is linked to poor patient outcomes after ARSI treatment in CRPC. To determine if the lineage plasticity risk signature was activated in primary tumors, we examined the TCGA dataset[38]. Importantly, only two of 495 patients had high risk scores (Supplementary

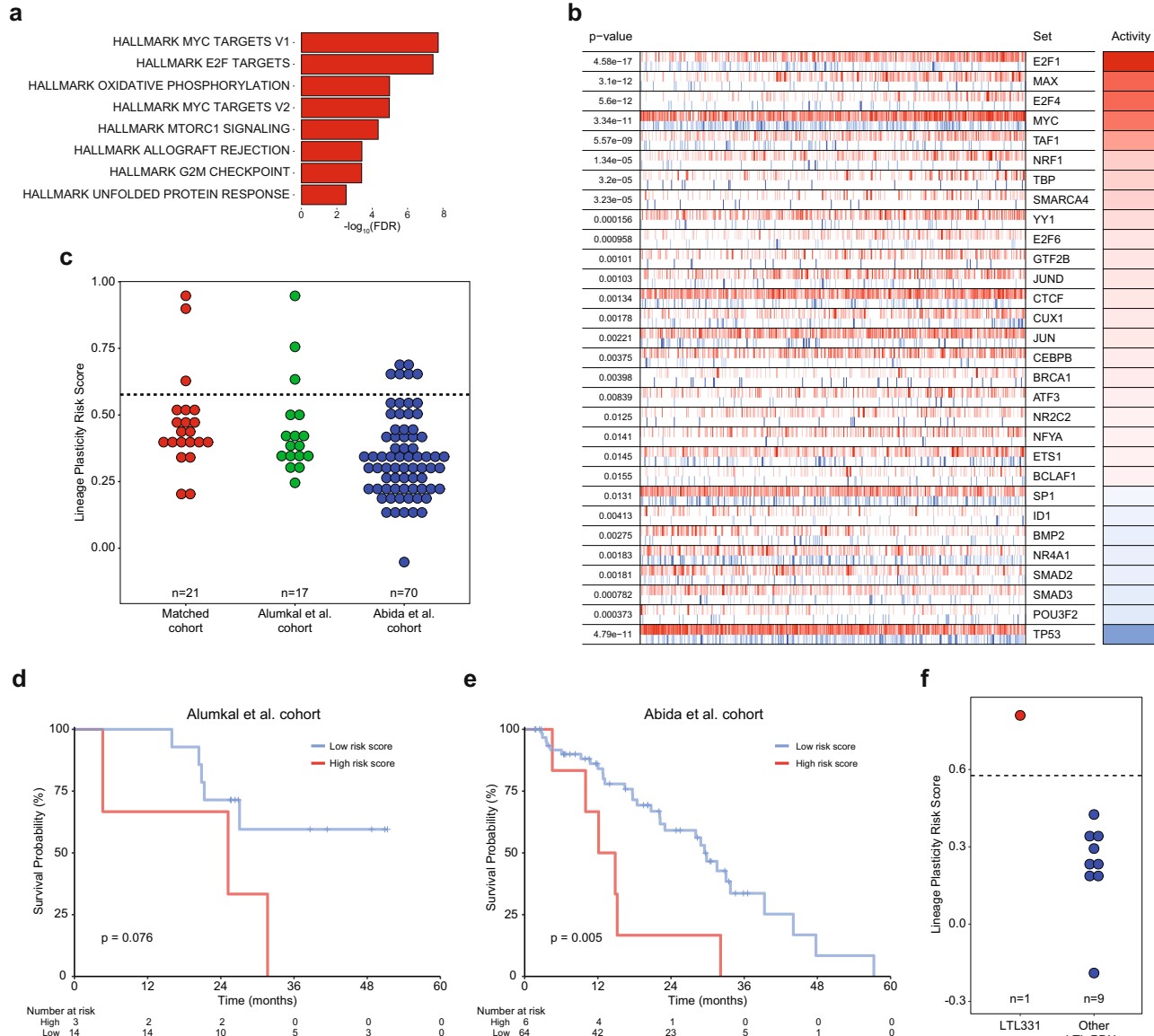

**Fig. 3 | Pathway and master regulator analysis implicate E2F1 in lineage plasticity risk, and a signature of lineage plasticity risk identifies tumors with poor outcomes after androgen receptor signaling inhibitor treatment. a** Hallmark pathway analysis of activated pathways in baseline samples for the three patients whose tumors converted (underwent lineage plasticity) vs. those patients whose tumors did not upon progression. **b** Master regulator analysis identifies top activated and deactivated transcription factors between converters and non-converters using the baseline tumor samples. Activity scores (right) and p-values (left, calculated using a gene shuffling test of the enrichment scores) were generated in the VIPER R package[26]. **c** Dot plot showing lineage plasticity signature score for patients in this cohort, the International Dream Team dataset described in Abida et al.[9] and unique patients not included in this matched biopsy cohort from Alumkal et al.[17]. **d, e** Kaplan-Meier survival curves for patients in the Alumkal et al. cohort (**d**) and Abida et al. cohort (**e**) stratified by high or low lineage plasticity risk score. p-values shown were determined using the log-rank test. **f** Dot plot showing lineage plasticity signature score for all castration naïve adenocarcinoma PDX models described by Lin et al.[22] Source data are provided as a Source Data file.

Fig. 2b). The lower frequency in primary tumors vs. CRPC cohorts suggests that activation of this lineage plasticity risk program may be induced by castration.

As stated previously, validation datasets with matched biopsies before and after ARSI treatment that include information on lineage at time of progression are lacking. However, previously we determined the impact of surgical castration on adenocarcinoma patient-derived xenografts (PDX)[22]. Nine PDXs do not undergo castration-induced lineage plasticity, while one PDX–LTL331–does and converts to a resistant version called LTL331R[22]. Importantly, the patient from whom the LTL331 PDX is derived had evidence of lineage plasticity in his tumor when it became castration-resistant, demonstrating this model's fidelity[22,39]. Our lineage plasticity risk signature was highly activated in LTL331 vs. the other hormone-naïve PDXs that do not undergo castration-induced lineage plasticity (Fig. 3f, Supplementary Fig. 2c, d). Indeed, LTL331 was the only PDX whose lineage plasticity risk score was greater than the cut-off defined in our matched biopsy cohort (Fig. 3f). Prior work demonstrates that the exome of LTL331 is strikingly similar to its castration-induced lineage plasticity derivative, strongly suggesting that transdifferentiation–rather than clonal selection–may explain conversion in this tumor[22]. Finally, the lineage plasticity risk score decreased in LTL331R vs. LTL331 (Supplementary Fig. 2c)–similar to the pattern we observed in the progression vs. baseline samples from converters in our matched biopsy cohort (Supplementary Fig. 2a).

**a**

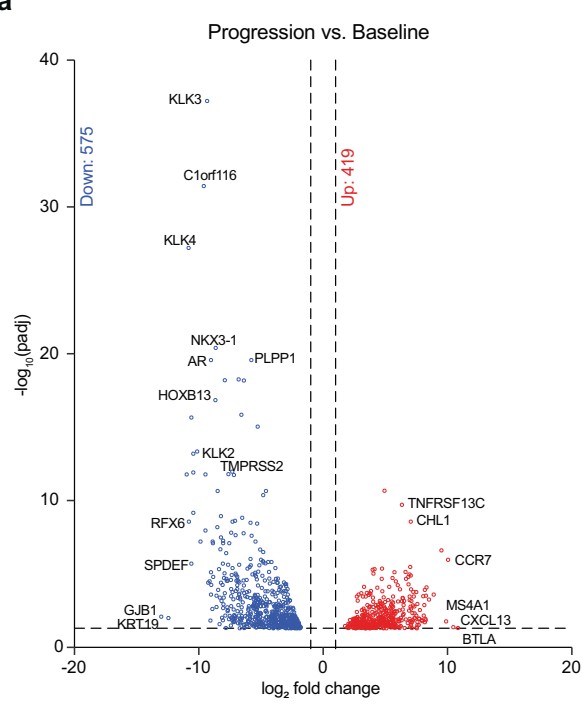

**b**

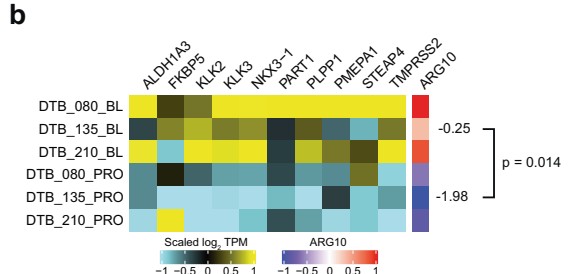

**c**

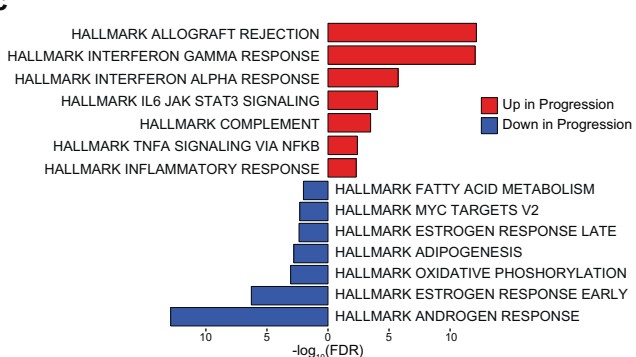

**Fig. 4 | Gene expression profiling identifies gene expression changes in tumors undergoing enzalutamide-induced lineage plasticity. a** Volcano plot showing top up and down regulated genes in progression samples vs. baseline samples for the three patients whose tumors converted (*n* = 3 pairs, no replicates). Adjusted p-values were calculated using the Wald test in the DESeq2 R package[55]. **b** ARG10 gene signature heatmap for three converters at baseline and progression. The left shows the expression levels of individual genes in the ARG10 signature, and the right shows the ARG10 signature score. p-value shown is for a two-tailed paired t-test between baseline and progression ARG10 scores (*n* = 3 pairs, no replicates). **c** Hallmark pathway analysis shows the top up or down regulated pathways in progression vs. baseline samples for the three patients whose tumors converted. Source data are provided as a Source Data file.

## Identification of transcriptional changes in tumors undergoing lineage plasticity

Next, we sought to understand changes induced by enza between the baseline and progression samples from the three converters more deeply. The top differentially expressed genes are shown in Fig. 4a. The *AR*, AR target genes (*KLK2, KLK3,* and *TMPRSS2*), and the AR coactivator *HOXB13* had markedly decreased expression (Fig. 4a, Supplementary Data 2). In keeping with this, progression biopsies from converters had significantly reduced expression of AR target genes from the ARG10 gene signature[27] (Fig. 4b). Though we found that genes from the Beltran NEPC Upregulated signature were increased in progression samples from converters (Supplementary Fig. 1d), it is worth noting that this signature contains both canonical NEPC genes and genes not explicitly associated with acquisition of neuroendocrine features that are AR-repressed. Specifically, examining canonical NEPC markers such as *SYP, CHGA,* and *NCAM1*, we found that these genes were not highly upregulated at progression (Supplementary Data 3). Importantly, other genes linked to NEPC (*SYT11, CIITA,* and *ETV5*)[21] or those normally repressed by the AR (*CDCA7L, FRMD3, IKZF3,* and *TNFAIP2*)[29] were more highly expressed in the progression biopsies, suggesting that these three converter tumors may be farther along the lineage plasticity spectrum than the previously described non-neuroendocrine DNPC subtype but not as far along as de novo NEPC or NEPC found at rapid autopsy by Labrecque et al.[23] that harbor a more complete neuroendocrine program.

Pathway analysis between baseline and progression samples from the three converters demonstrated enrichment in several pathways, including: allograft rejection, interferon gamma response, interferon alpha response, and IL6/JAK/STAT signaling (Fig. 4c). Conversely, androgen and estrogen response—both linked to luminal differentiation—were the most downregulated, confirming loss of AR-dependence. We examined differences in gene expression between baseline and progression samples from the 18 patients whose tumors did not undergo lineage plasticity. Several of the pathways activated in the converter tumors were also activated in the non-converters—namely, interferon alpha response, interferon gamma response, and TNF-α signaling (Supplementary Fig. 3). Uniquely upregulated pathways in the converters include: allograft rejection, IL6-JAK-STAT3 signaling, inflammatory response, and complement. Uniquely downregulated pathways in the progression samples from non-converters included: E2F targets, G2M checkpoint, and hedgehog signaling. The only uniquely upregulated pathway in non-converters was protein secretion, while uniquely downregulated pathways included hedgehog signaling, G2M checkpoint, and E2F targets.

## Protein expression analysis demonstrates switch to double negative prostate cancer in samples undergoing lineage plasticity

To understand the architecture of the tumors from the three converters, we used multiplex immunofluorescence (IF) with three luminal lineage markers (AR, NKX3.1, and HOXB13)—all downregulated at the mRNA level by RNA-sequencing (Fig. 4a)—and the NEPC marker INSM1[40]. LuCaP PDX samples were used as positive and negative controls (Supplementary Fig. 4a). Matched tissue samples for multiplex IF were available for subjects 135 and 210 but not for subject 80 (Fig. 5). We identified one additional WCDT subject (103) with matched biopsies whose tumor underwent rapid clinical progression after enza treatment in the setting of a falling serum PSA—a clinical marker of AR-independence. Matched RNA-sequencing was not available for this subject, but his tumor exhibited evidence of lineage plasticity (Fig. 5). There was a spectrum of AR, NKX3.1, and HOXB13 expression in baseline samples with some cells expressing low levels of each marker,

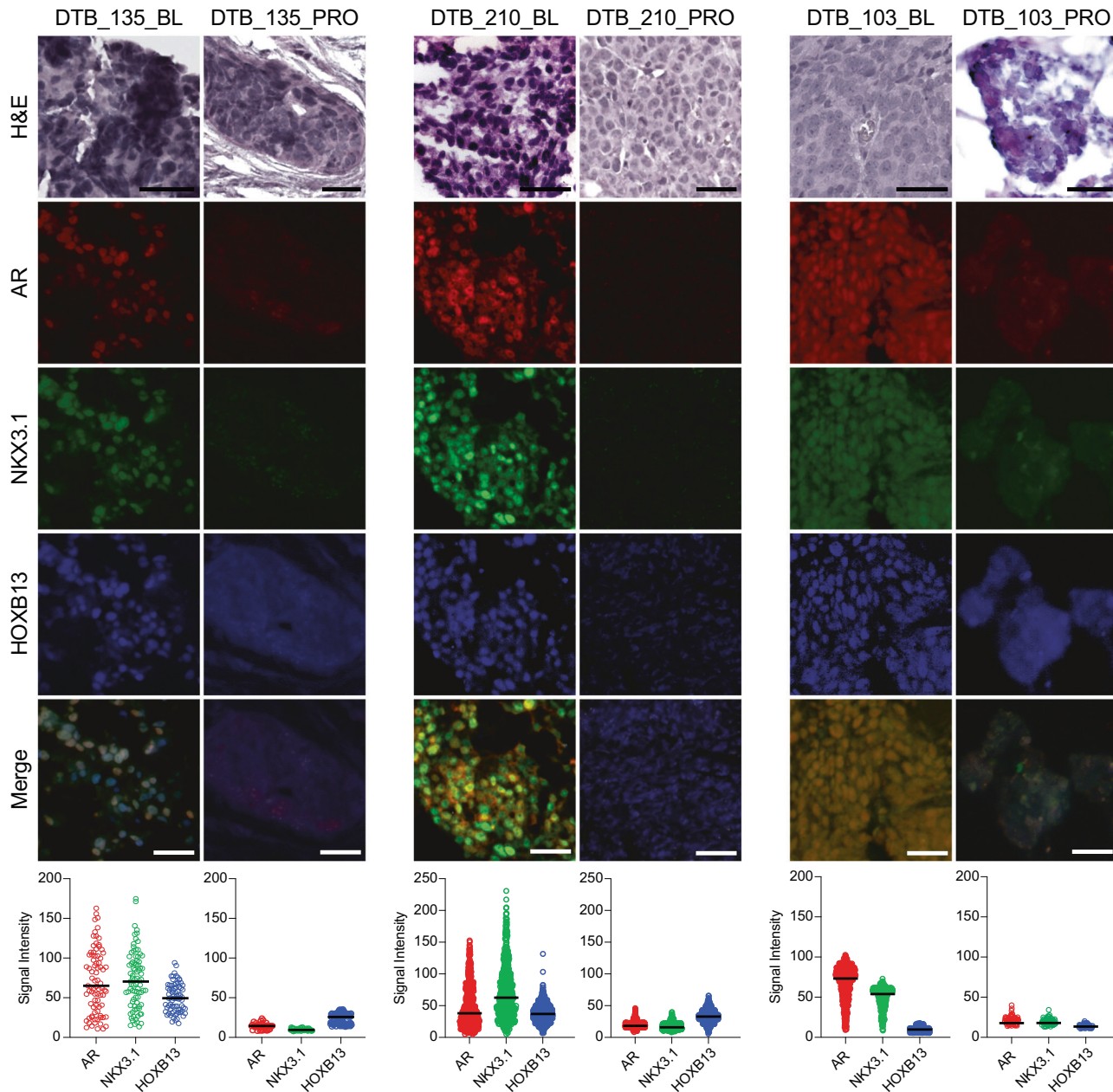

**Fig. 5 | Multiplex immunofluorescence demonstrates switch to double negative prostate cancer in samples undergoing lineage plasticity.** Patients 135, 210, and an additional West Coast Dream Team patient whose tumor converted, patient 103, were stained for AR, NKX3.1, and HOXB13 expression ($n = 6$ biologically independent samples with no replicates). The scale bar represents 50 μm. Signal intensity values for each marker are shown with the median value indicated. Signal intensity values were compared between each matched pair using the Mann-Whitney two-tailed test with $p < 1 \times 10^{-15}$ for each comparison except DTB_103 HOXB13, which is $p = 1.8 \times 10^{-14}$. Source data are provided as a Source Data file.

while other cells expressed higher levels. However, at progression, there was a convergence towards population-wide loss of AR, NKX3.1, and HOXB13 in each sample. We did not identify INSM1 upregulation in any of the baseline or progression tumors (Supplementary Fig. 4b). These results match our RNA-sequencing that failed to demonstrate upregulation of other canonical NEPC markers (Supplementary Data 3) and that characterized the three converter samples as DNPC by the Labrecque classifier, rather than NEPC (Fig. 2b).

### Conservation of DNA mutations in tumors undergoing lineage plasticity

Finally, to determine if the progression samples from converters represented distinct clones with unique genetic alterations vs. baseline, we performed DNA mutation and copy number analysis. For subjects 80 and 103, the same tumor lesion was biopsied at baseline and progression. DNA-sequencing of these biopsies showed identical DNA mutations. For subjects 135 and 210, matched metastatic biopsy DNA-sequencing was unavailable. However, cell-free DNA was available. DNA-sequencing of cell-free DNA samples showed that mutations and copy number alterations were conserved between baseline and progression samples (Table 1).

Loss of the tumor suppressor genes *TP53, RB1*, and *PTEN* has been linked to lineage plasticity risk in pre-clinical models[31,32]. However, we do not know if the presence of these genomic abnormalities in patient tumors is associated with the risk of lineage plasticity to DNPC. One of the three converter patients (subject 80) was found to have an inactivating *PTEN* mutation and a second patient (subject 103) had *RB1 loss*, but none were found to have compound *TP53/RB1/PTEN* loss. When

**Table 1 | DNA sequencing of matched samples from converters demonstrates conserved alterations**

| Patient ID | Mutation | Copy number gain/loss |
|---|---|---|
| DTB_80_BL | PTEN | |
| DTB_80_Pro | PTEN | |
| DTB_103_BL | RB1, FGFR3, NOTCH1 | |
| DTB_103_Pro | RB1, FGFR3, NOTCH1 | |
| DTB_135_BL | SPEN, FAT1 | AR amplification, MYC amplification |
| DTB_135_Pro | SPEN, FAT1, CTNNB1 (subclonal) | AR amplification, MYC amplification |
| DTB_210_BL | APC, SPOP, KMT2C | |
| DTB_210_Pro | APC, SPOP, KMT2C | |

available, we also examined *TP53/RB1/PTEN* status for tumors from the Abida et al.[9] and Alumkal et al.[17] cohorts that had high lineage plasticity risk scores. Of the seven high lineage plasticity risk score tumors examined from these two validation cohorts, only two tumors had loss of two or more of the genes *TP53, RB1, and PTEN* (Supplementary Table 4). DNA-sequencing of matched metastatic biopsies for the cohort as a whole is shown in Supplementary Table 5.

## Discussion

Despite recent advances in treatment, resistance to ARSIs like enza remains a significant medical issue. This report represents a unique collection of matched patient metastatic biopsies with RNA-sequencing before treatment with enza and upon progression, providing a useful resource to understand mechanisms that contribute to clinical enza resistance.

We determined that most progression tumors clustered with their baseline pair and that most tumors did not change their transcriptional cluster designation[23,24] between baseline and progression. Many patients' matched tumors expressed a similar gene expression program, regardless of whether a different lesion or tissue type was biopsied, suggesting homogeneity of many lesions within the same patient.

Maintenance of AR function has been identified as a common mechanism of ARSI resistance due to mutations, splice variants, and amplifications of the *AR* gene or an upstream *AR* enhancer[12,41]. In our cohort, we found that AR activity was unchanged or increased in nearly one-half of patients even though patients were still taking enza at the time of progression biopsy, demonstrating the continued reliance of tumors on the AR, which matches prior, smaller reports[12,14]. Conversely, in one-half of patients, AR activity was reduced at progression, suggesting other AR-independent resistance mechanisms in these tumors or activation of a non-canonical AR program.

Tumors that undergo lineage plasticity after enza appear to be quite aggressive[24], but predictors of risk for this form of resistance are poorly defined. We identified three baseline tumors that may have been in an intermediate state at high risk of lineage plasticity. In each case, these patients' progression biopsies underwent significant transcriptional changes—changing from ARPC to DNPC/Aggarwal cluster 2 at progression. Orthogonal measurements of AR loss and lineage plasticity confirmed the conversion of these three tumor samples. Pathway analysis of progression vs. baseline biopsies from the converters demonstrated upregulation of several inflammatory pathways found previously to be activated in cancer stem cells[42,43]—interferon alpha, interferon gamma response, allograft rejection, IL6/JAK/STAT signaling, and TNF-alpha signaling via NFKB. Interestingly, these pathways were found to be associated with de novo clinical enza resistance in our prior work[17]. Several of these same pathways—interferon alpha, interferon gamma response, and TNF-alpha signaling via NFKB—were also significantly activated in pathway analysis of progression vs. baseline biopsies from non-converters. These results

suggest that enza treatment may contribute to a more stem-like state—including in non-converter tumors that did not yet have evidence of lineage plasticity at the timepoint examined—that contributes to resistance. Future studies targeting these pathways may clarify their importance for promoting enza resistance.

Each of the three converter samples had an increase in Beltran NEPC signature scores[21] at progression, but they did not express canonical NEPC markers such as *SYP, CHGA*, or *INSM1*. Instead, these tumors were chiefly defined by their loss of AR signaling, most compatible with DNPC[18]. Recent work by Taavitsainen et al. and Han et al. demonstrates that enzalutamide induces transcriptional reprograming of prostate cancer models—an effect that is partially mediated by changes in chromatin structure[44,45]. Importantly, we found that DNA mutations and copy number alterations were conserved between samples from these converters. These results suggest that the marked differences in gene expression observed and transcriptional conversion in these tumors may be epigenetically regulated, rather than due to selection of an independent, genetically distinct clone that is different from the baseline bulk tumor population.

We identified a signature of 14 genes highly upregulated in baseline tumors from the converters. We are not aware of any published datasets that include matched biopsies with RNA-sequencing and information on change in tumor phenotype. Therefore, it was not possible to validate the predictive ability of this signature of lineage plasticity risk. However, high activity of this signature in two independent cohorts[9,17] was associated with poor overall survival after ARSI treatment. Because progression biopsies did not exist, we may only speculate whether lineage plasticity was responsible for these patients' poor outcomes. That this lineage plasticity risk signature was strongly enriched in the LTL331 PDX that undergoes castration-induced lineage plasticity vs. other adenocarcinoma PDXs that do not[22] suggests that our signature may in fact be measuring risk of susceptibility to castration or ARSI-induced lineage plasticity. Importantly, this signature decreased in the progression biopsies from all three converters and in the LTL331R PDX, suggesting this signature may be measuring the capacity of tumor cells to undergo lineage plasticity—rather than identifying tumors that have already undergone lineage plasticity.

Loss of the tumor suppressor genes *TP53, RB1*, and *PTEN* has been linked to NEPC lineage plasticity risk in pre-clinical models[31,32]. Importantly, our three converter tumors underwent lineage switching to DNPC, rather than NEPC. TP53 was inferred to be the most deactivated master regulator in the baseline biopsies from the converters, and we determined that an *RB1* loss signature was increased at progression in converters. Two out of four of the tumors from the converter patients in our matched biopsy cohort harbored genetic alterations in at least one of these genes by the assays used in this paired analysis. These genes can also be lost via structural alterations that we did not measure in our DNA sequencing assay or via non-genetic mechanisms[30,46]. In fact, we previously performed whole genome sequencing on the baseline tumor from one of the converters (subject 80) and found a structural alteration in *TP53* using that approach[41]. However, in examining the *TP53, RB1, and PTEN* status for tumors from the Abida et al.[9] and Alumkal et al.[17] cohorts, we determined that only two of the seven tumors with high lineage plasticity risk scores had loss of two or more of these genes. In the future, it will be important to determine whether combining *TP53, RB1, and PTEN* status with the gene signature we identified is better than either alone for identifying tumors at risk of lineage plasticity.

The conservation of genetic alterations across each converter patient's matched sample suggests that epigenetically regulated adaptive changes may explain lineage plasticity in these cases. It is striking that the baseline and progression sample from one patient (subject 135) whose tumor underwent conversion had *AR* amplification despite loss of *AR* expression at progression. These data strongly suggest that the progression sample underwent lineage plasticity from a baseline tumor

population that was once AR-driven. There are several possible explanations for loss of *AR* expression despite the *AR* amplification, including repressive histone methylation or DNA methylation[31,47].

Multiplex IF showed heterogenous expression of AR, NKX3.1, and HOXB13 at baseline in the converter patients. At progression, all these genes were subsequently lost. We cannot entirely rule out that the low AR/NKX3.1/HOXB13 tumor cells at baseline were selected for by enza treatment. However, that the signature of lineage plasticity risk we identified in the baseline samples declined—rather than increased—at progression suggests that enza changed the transcriptome in the tumor population, rather than selecting for outgrowths of AR-low tumors that harbor the same starting program at baseline and progression. Further, we did not find a dominant, mutationally distinct clone at progression vs. baseline for any of the four converters we examined. This strongly suggests that selection of a genetically unrelated clone distinct from the baseline bulk tumor population did not occur.

Pathway analysis in baseline tumors from converters vs. non-converters revealed upregulation of MYC and E2F targets, both of which have been implicated in NEPC lineage plasticity previously[24,29,31,48]. Additionally, E2F1 was inferred to be the top differentially activated master regulator in comparing these samples. Our previous work identified BET bromodomain inhibition as a promising therapeutic strategy to target AR-low, E2F1-high tumors[29]. Additionally, our prior phase II clinical trial of the BET bromodomain inhibitor ZEN-3694 determined that AR-low, E2F1-high tumors may be particularly susceptible to BET bromodomain inhibition[29,49]. These results suggest that BET bromodomain inhibition is worthy of further study as a clinical strategy to target tumors at risk of lineage plasticity or those that have already undergone lineage plasticity.

The study has several strengths. It was carried out prospectively and is a valuable CRPC cohort, containing matched biopsies before and after enza treatment for which transcriptional profiling and tumor phenotype data are available. This study also has limitations. First, we cannot confirm whether tumors from patients with high expression of the lineage plasticity risk signature in the independent cohorts we examined underwent lineage plasticity after ARSI treatment because progression biopsy information was unavailable. Second, though our studies with multiplex IF, DNA-sequencing, and laser capture microdissected RNA-sequencing suggest that epigenetic changes may explain why the converter tumors underwent lineage plasticity, our studies did not include a detailed examination of chromatin accessibility or conformation—critical determinants of cell state and the capacity for cell state changes—because sufficient tumor material was unavailable. Finally, progression-free survival in our cohort is shorter than previous studies in patients treated with enza[6,25,50]. Patients included in this current study had lesions that were amenable to biopsy at baseline and progression and sufficient tumor tissue found on biopsy to perform RNA-sequencing. Thus, it is quite likely that our cohort was enriched for patients with more aggressive disease.

In summary, our study demonstrates the importance of transcriptional profiling to understand clinical enza resistance. In the future, identifying biomarkers for those whose tumors are at greatest risk of undergoing lineage plasticity vs. those whose tumors are most likely to remain AR-driven may facilitate development of up-front combination clinical trials testing drugs predicted to block these two principal enza resistance mechanisms.

## Methods

### West Coast Dream Team (WCDT) metastatic tissue collection
All biopsies, data collection, and analyses were performed in compliance with all relevant ethical regulations after informed consent under an IRB-approved protocol (NCT02432001) at the participating WCDT centers' (University of California, Davis; University of California, Los Angeles; University of California, San Francisco; Oregon Health & Sciences University; and University of British Columbia) IRBs.

Metastatic tissue was collected by computed tomography or ultrasound-guided biopsies in accordance with the standard operating procedure and institutional standards, with the goal of minimizing patient risk. Biopsies were snap frozen. RNA-sequencing was performed on matched, paired biopsies from 21 men ages 58-88 (median 71) with metastatic, castration-resistant prostate cancer who had a tissue biopsy performed prior to starting treatment with enza and a second biopsy performed at time of progression. Primary outcome measure for this protocol: proportion of mCRPC patients with high androgen receptor activity determined by a gene-expression-based signature for Androgen Receptor activity having a probability of >0.50. Secondary outcome measure: progression free survival and overall survival measured from the start of therapy after the baseline biopsy until progression. Data were collected from the University of California, Los Angeles; University of California, Davis; University of California, San Francisco; Oregon Health & Science University; and the University of British Columbia between May 1, 2015 and October 8, 2021. Compensation was not provided to study participants.

### RNA-sequencing and data processing
Core biopsy samples were flash frozen in Optical Cutting Temperature (OCT) for gene expression analysis. Laser capture microdissection was performed on frozen sections to enrich for tumor content[51]. Total RNA was isolated (Stratagene Absolutely RNA Nano Prep) (RIN > 8) and amplified using NuGEN Ovation RNA seq System V2. Libraries were generated using NuGEN Ovation Ultralow System V2 for Illumina sequencing. RNA seq was performed on the Illumina NextSeq 500, PE75 with at least 100 M read pairs. The raw fastq files were first quality checked using FastQC (version 0.11.8) software (http://www.bioinformatics.bbsrc.ac.uk/projects/fastqc/). Fastq files were aligned to hg38 human reference genome and per-gene counts and transcripts per million (TPM) quantified by RSEM[52] (version 1.3.1) based on the gene annotation gencode.v28.annotation.gtf.

### Unsupervised clustering
To understand the overall transcriptional similarities across these 21 paired samples, unsupervised clustering was performed using RNA-sequencing data. Briefly, the raw count matrix was filtered to remove low expression genes and genes with raw count >= 20 in at least two samples were kept. The filtered count matrix was transformed using the *vst* function implemented in DESeq2 R package (version 1.22.2)[53]. The transformed values were used to compute the sample-to-sample Euclidean distance metric for hierarchical clustering through the 'complete' method. To cluster samples prior to treatment (baseline), TPM gene expression data was first filtered to remove low expression genes as described above and non-protein-coding genes as annotated by HUGO Gene Nomenclature Committee (HGNC). The filtered TPM matrix was log transformed, and the 500 or 1000 most varying genes were selected to compute the sample-to-sample gene expression spearman correlation which was then converted to distance followed by clustering through the 'complete-linkage hierarchical clustering' method.

### Differential expression gene, pathway, and master regulator analysis
Differential gene expression analysis was performed using DESeq2 (version 1.22.2). Gene expression differences were considered significant if passing the following criteria: adjusted P-value < 0.05, absolute fold change ≥ 1.5. For the converter vs non-converter baseline sample comparison, we used the adjusted P-value < 0.1. The Wald test statistics from DESeq2 output was used as pre-ranked gene list scores to perform pathway analysis using *cameraPR* implemented in limma R package (version 3.38.3)[54] and the hallmark collection from MSigDB database (version 7.0). Transcription factor activity was inferred using the master regulator inference algorithm[55] (MARINa) implemented in the VIPER R package (version 1.16.0)[26]. Pre-ranked gene list scores and

a regulatory network (regulome) are the two sources of data required as input for VIPER analysis. The pre-ranked gene list scores were the same as above and the transcription factor regulome used in this study was curated from several databases as previously described[56].

### Single sample AR activity

To measure single-sample AR regulon activity, we used the VIPER R package (version 1.16.0)[26] with the log2 transformed TPM gene expression matrix as input. The regulon used in VIPER analysis was the same as described above. Scores were considered to have marked difference if change between baseline and progression sample was ≥20% of the range between all samples.

### Multiplex immunofluorescence

Multiplex immunofluorescence studies using AR (Cell signaling Technologies, 5153 T, rabbit monoclonal clone D6F11, 1:100 dilution), INSM1 (Santa Cruz, sc-271408, mouse monoclonal clone A-8, 1:50 dilution), NKX3.1 (Fisher, 82788, rabbit polyclonal, 1:50 dilution) and HOXB13 (Cell signaling Technologies, 90944 S, rabbit monoclonal clone D7N8O, 1:200 dilution) antibodies were carried out on archival formalin fixed paraffin embedded (FFPE) tissues. In brief, 5 μM paraffin sections were de-waxed and rehydrated following standard protocols. The staining protocol consisted of four sequential staining steps, each with tyramide-based signal amplification using the Tyramide SuperBoost kits (Thermo Fisher) as described previously[57]. De-waxed slides were first subjected to steaming for 40 min in Target Retrieval Solution (Agilent, S1700) and incubated with AR specific antibodies (1:100). Signal amplification was carried out by first incubating slides with PowerVision Poly-HRP anti-rabbit (Leica, PV6119, no dilution) secondary antibodies followed by Tyramide568 (Tyramide SuperBoost kit, Thermo Fisher) according to manufacturer's protocols. Slides were then stripped by steaming in citrate buffer (Vector) for 20 minutes and subsequently incubated with INSM1 specific antibodies (1:50) followed by PowerVision Poly-HRP anti-mouse (Leica, PV6114, no dilution) secondary antibodies and Tyramide647 (Tyramide SuperBoost kit). Next, slides were stripped for 20 min in Target Retrieval Solution (S1700, Agilent), incubated with NKX3.1 specific antibodies (1:50) followed by PowerVision Poly-HRP anti-rabbit (Leica, PV6119, no dilution) secondary antibodies and Tyramide488 (Tyramide SuperBoost kit). Lastly, slides were steamed in Citrate buffer (Vector) for 20 min, incubated with HOXB13 antibodies (1:200) followed by PowerVision Poly-HRP anti-rabbit (Leica, PV6119, no dilution) secondary antibodies and Tyramide350 (Tyramide SuperBoost kit). Slides were mounted with Prolong (Thermo Fisher), imaged on a Nikon Eclipse E800 (Nikon) microscope, and image analyses were carried out using QuPath (version 0.3.0)[58]. Regions of interest containing cancer cells were outlined using the polygon annotation tool, and the positive cell detection feature was used to segment individual cells in all channels. The mean nuclear intensity was extracted for each single cell using the detection measurements function for each channel separately, and statistical tests were performed using GraphPad Prism (version 9.3.1).

### DNA-sequencing

Next generation targeted genomic DNA-sequencing of FFPE tissue was performed using a 124 gene as previously described[59]. Cell-free DNA was extracted from approximately 1 mL of previously banked plasma and subjected to low-pass whole-genome-sequencing (WGS) and targeted deep sequencing using the Ion TorrentTM Next-Generation Sequencing (NGS) system (Thermo Fisher Scientific, Waltham, MA), as described previously[60]. NGS reads were processed using Ion Torrent SuiteTM and analyzed with standard workflows in Ion ReporterTM (Thermo Fisher Scientific) and established in-house bioinformatics pipelines. Tumor content estimates were derived from low-pass WGS data using the ichorCNA package (version 0.3.2) in R[61]. Total mapped NGS reads for low-pass WGS ranged from 4,235,342–6,185,948 (corresponding to

0.202–0.292× coverage). Targeted deep sequencing was performed using the OncomineTM Comprehensive Assay Plus (Thermo Fisher Scientific), which targets greater than 1 Mb of genomic sequence corresponding to more than 500 genes recurrently altered in human cancers; total mapped NGS reads for targeted sequencing ranged from 5,069,230–8,497,096 (corresponding to 347–596× coverage across the targeted regions). Prioritized variants and copy number alterations from targeted NGS data were manually curated by an experienced molecular pathologist (A.M.U.) using previously established criteria[62].

### Statistics & reproducibility

Descriptive statistics were performed using R (version 1.16.0) and GraphPad Prism (version 9.3.1). No statistical method was used to predetermine sample size. No data were excluded from the analyses. The experiments were not randomized. All patients from the Stand Up to Cancer Foundation/Prostate Cancer Foundation West Coast Dream Team who underwent a metastatic tumor biopsy prior to enza and a repeat biopsy at the time of progression and whose tumor cells underwent RNA-sequencing after laser capture microdissection were included. The investigators were not blinded to allocation during experiments and outcome assessment. Single samples were used in all cases as replicates were not possible.

### Reporting summary

Further information on research design is available in the Nature Research Reporting Summary linked to this article.

### Data availability

Source data are provided with this paper. The raw RNA-seq data generated in this study are available under restricted access because they contain clinical information; access can be obtained by receiving permission from study authors to access the dataset on EGA. The raw RNA-seq data used in this study are available in the EGA database under Study ID EGAS00001005954. The dataset is limited to academic use only. Requests for access to the dataset should be directed to Eric J. Small (eric.small@ucsf.edu). All requests will be reviewed within 15 working days. Additionally, the RNA-seq data (TPM) generated in this study are provided as Supplementary Data 4.

Details of experimental procedures for other methods, including RNA-seq data analysis, gene signature analysis, and survival analysis are included in Supplementary Methods. Source data are provided in this paper.

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

## Acknowledgements

This research was supported by the Stand Up to Cancer-Prostate Cancer Foundation (PCF) Prostate Dream Team Translational Cancer Research Grant SU2C-AACR-DT0409 (T.M.B., R.A., D.A.Q., C.J.R., M.G., J.H., C.P.E., P.L., R.E.R., O.W., M.R., J.M.S., G.V.T., F.Y.F., E.J.S., J.J.A.) and a Prostate Cancer Foundation Challenge Award (J.J.A.). Other support includes: National Cancer Institute (NCI) R01 CA251245 (J.J.A.), R01 CA234715 (PSN); The Pacific Northwest Prostate Cancer Specialized Programs of Research Excellence (SPORE) NCI P50 CA097186 (M.C.H., C.M., P.S.N., J.J.A.); the Michigan Prostate SPORE NCI P50 CA186786 (AMU, J.J.A.); NCI P01 CA163227 (PSN); The Drug Resistance and Sensitivity Network NCI U54 CA224079 (PSN) and NCI P50 CA186786-07S1 (J.J.A.); NCI T32 CA009357 (T.C.W.); University of Michigan Rogel Cancer Center Innovation Award NCI P30 CA046592 (J.J.A.); Department of Defense Idea Award (W81XWH-20-1-0405) (J.J.A.); Department of Defense Idea Award (W81XWH2110539) (Z.X.); Grant 2021184 from the Doris Duke Charitable Foundation (M.C.H.); the V Foundation (MCH); National Comprehensive Cancer Network (NCCN)/Astellas Pharma Global Development Award (J.J.A.); the Sheppard Family Fund (J.J.A.). We would like to thank the staff of the Rogel Cancer Center Liquid Biopsy Shared Resource for technical expertise in isolation and molecular profiling of plasma cell-free DNA

## Author contributions

Conceptualization, J.J.A., Data curation, T.C.W., X.G., C.J.L., A.M.U., R.A., F.Y.F., E.J.S., C.K.W., J.M.S., Z.X., J.J.A., Formal analysis, T.C.W., X.G., D.F., C.J.L., A.M.U., M.C.H., Y.H., D.S., I.C., C.K.W., A.S.W., V.U., J.M.S., J.A.Y., Z.X., J.J.A., Funding acquisition, E.J.S., O.W., J.J.A., Investigation, T.C.W., X.G., E.R., D.F., C.J.L., A.M.U., R.A.P., M.C.H., Y.H., D.S., T.M.B., A.F., R.A., F.Y.F., D.A.Q., E.J.S., J.F.Y., C.J.R., M.G., Y.W., J.H., I.C., C.M., P.S.N., C.P.E., P.L., R.E.R., O.W., M.R., C.K.W., A.S.W., V.U., J.M.S., G.V.T., J.A.Y., Z.X., J.J.A., Methodology, N/A, Resources, T.M.B., R.A., E.J.S., C.J.R., M.G., Y.W., J.H., C.P.E., P.L., R.E.R., O.W., M.R., G.V.T., J.J.A., Software, N/A, Supervision, Z.X., J.J.A., Validation, N/A, Visualization, T.C.W., X.G., D.F., M.C.H., Y.H., J.A.Y., Z.X., J.J.A., Writing original draft, T.C.W., J.J.A., Writing review, T.C.W., X.G., E.R., D.F., C.J.L., A.M.U., R.A.P., M.C.H., Y.H., D.S., T.M.B., A.F., R.A., F.Y.F., D.A.Q., E.J.S., J.F.Y., C.J.R., M.G., Y.W., J.H., I.C., C.M., P.S.N., C.P.E., P.L., R.E.R., O.W., M.R., C.K.W., A.S.W., V.U., J.M.S., G.V.T., J.A.Y., Z.X., J.J.A.

## Competing interests

O. Witte currently has consulting, equity, and/or board relationships with Trethera Corporation, Kronos Biosciences, Sofie Biosciences, Breakthrough Properties, Vida Ventures, Nammi Therapeutics, Two River, Iconovir, Appia BioSciences, Neogene Therapeutics, 76Bio, and Allogene Therapeutics outside of submitted work. T.M. Beer reports consulting fees from AbbVie, Arvinas, Astellas Pharma, AstraZeneca, Bayer, Constellation, Grail Inc., Janssen, Myovant Sciences, Pfizer, Sanofi, Sapience Therapeutics, Bristol-Myers Squib, Novartis, Clovis Oncology, stock ownership in Arvinas Inc, Salarius Pharmaceuticals and researching funding from Alliance Foundation Trials, Astellas Pharma, Bayer, Boehringer Ingelheim, Corcept Therapeutics, Endocyte Inc./Advanced Accelerator Applications (AAA), Freenome, Grail Inc., Harpoon Therapeutics, Janssen Research & Development, Medivation Inc., Sotio, Theraclone Sciences/OncoResponse, Zenith Epigenetics, all outside of submitted work. J. Alumkal has received consulting and speaker's fees from Astellas Pharma, consulting fees from Dendreon, consulting fees from Merck, consulting fees from Bristol Myers Squibb, and research support to his institution from Astellas Pharma, Zenith Epigenetics, and Beactica. The remaining authors declare no conflicts of interest.

## Additional information

**Supplementary information** The online version contains

supplementary material available at https://doi.org/10.1038/s41467-022-32701-6.

[1]Division of Hematology and Oncology, Department of Internal Medicine, Rogel Cancer Center, University of Michigan, Ann Arbor, MI, USA. [2]Knight Cancer Institute, Oregon Health & Science University, Portland, OR, USA. [3]Department of Pathology, Michigan Center for Translational Pathology, Rogel Cancer Center, University of Michigan, Ann Arbor, MI, USA. [4]Divisions of Human Biology and Clinical Research, Fred Hutchinson Cancer Research Center, Seattle, WA, USA. [5]Helen Diller Family Comprehensive Cancer Center, University of California San Francisco, San Francisco, CA, USA. [6]Department of Medicine, University of California San Francisco, San Francisco, CA, USA. [7]Masonic Cancer Center, University of Minnesota; Department of Laboratory Medicine and Pathology, University of Minnesota, Minneapolis, MN, USA. [8]Department of Urological Sciences and Vancouver Prostate Centre, University of British Columbia, Vancouver, BC, Canada. [9]Department of Experimental Therapeutics, BC Cancer, University of British Columbia, Vancouver, BC, Canada. [10]Duke University, Durham, NC, USA. [11]Department of Urology, University of Washington, Seattle, WA, USA. [12]University of California Davis, Davis, CA, USA. [13]University of California Los Angeles, Los Angeles, CA, USA. [14]Department of Microbiology, Immunology, and Molecular Genetics at the David Geffen School of Medicine, UCLA, Los Angeles, CA, USA. [15]VA Greater Los Angeles Healthcare System, Los Angeles, CA, USA. [16]UC Santa Cruz Genomics Institute and Department of Biomolecular Engineering, University of California, Santa Cruz, Santa Cruz, CA, USA. [17]Departments of Radiation Oncology and Urology, University of California San Francisco, San Francisco, CA, USA. [18]These authors contributed equally: Thomas C. Westbrook, Xiangnan Guan. [19]These authors jointly supervised this work: Zheng Xia, Joshi J. Alumkal. ✉e-mail: xiaz@ohsu.edu; jalumkal@med.umich.edu

