## [Peer Review File · Nature Communications]

Transcriptional Profiling of Matched Patient Biopsies Clarifies Molecular Determinants of Enzalutamide-Induced Lineage PlasticityREVIEWER COMMENTS

Reviewer #1 (Remarks to the Author): expert in prostate cancer

In this well-written manuscript by Westbrook et al, the authors address an important question: do we see evidence of lineage plasticity via expression analysis in prostate cancers that are progressing on the next-generation AR inhibitor enzalutamide? The authors are to be commended for performing bulk RNA sequencing on microdissected tumors at baseline and at progression from 21 patients (same tumor biopsied for 8/21 patients) – this is no easy task and is the largest dataset of its kind. One of the most interesting findings is a negative one: most tumors (18/21) did not show a substantial change of transcriptional profile at progression, and half maintained high AR profile which is consistent with clinical observations: most patients with prostate cancer still die of high PSA producing disease. Nonetheless, lineage plasticity remains important biologically and clinically given its associated aggressive clinical behavior. 3/21 patients, called ‘converters’ had evidence of transformation to a lineage plasticity phenotype based on change in their expression profiles at progression. The authors characterize transcriptional changes for those 3 patients and derive an exploratory signature from their baseline tumors that may be associated with risk of developing a lineage plasticity phenotype.

I believe this is important novel work that merits publication with some minor comments:

1- In supplementary data, the authors should provide patient level clinical and genomic information in table format (ie sample type, enzalutamide response characteristics, prior treatment and most importantly RNAseq results and DNA sequencing when available). The summary tables do not provide sufficient detail.

2- As the authors note in the discussion, overall enzalutamide response in this 21 patient cohort is generally poor despite relatively small proportion of patients receiving prior abiraterone or taxane therapy. Suggest commenting on this in the results section too.

3- In the Fig 1 clustering analysis, were the ‘closest’ match pairs the cases where the same site was biopsied? Did tumors cluster together based on site of metastasis? Curious to know if microdissection mostly removes transcriptional effect of microenvironment.

4- Lines 178-179: authors suggest that the conversion rate seen here (3/21 = 14%) nearly matches the frequency of 17% reported in Aggarwal et al JCO. I do not think this is a good comparison: Aggarwal et al focused on IHC to define treatment emergent NEPC and found a surprisingly high rate of 17%. As the authors state here, the 3 converters did not show transcriptional evidence of NEPC and no samples displayed significant NEPC features at progression despite the aggressive behavior of this cohort. The authors should modify this statement and comment on the absence of NEPC features in their cohort.

5- The authors derive a 14 gene expression signature from the (only) 3 converters which they suggest is associated with increased risk of lineage plasticity. They show that this signature is associated with worse outcomes in independent cohorts, and that it is associated with lineage plasticity in a PDX model. Evidently this is very exploratory, but to try to address the potential future utility of such a signature, can the authors comment in the discussion on how such a signature may be used in the context of other molecular profiles e.g. TP53/RB1/PTEN status, and how a marker of lineage plasticity risk may be clinically useful?

6- Along the same line, it is surprising to see that none of the converters have compound tumor suppressor genomic alterations (Table 1) that so far are considered to be the strongest markers of developing lineage plasticity. Can the authors comments on mutational status in TP53/RB1/PTEN in the other tumors in their cohort, as well as on mutational status in the signature positive cases in the independent cohorts (Fig3D-E)?

Reviewer #2 (Remarks to the Author): expert in prostate cancer biomarkers

This is an interesting manuscript in which the authors performed laser capture microdissection and RNA-seq on matched metastatic tumor biopsies prior to enzalutamide and at the time of progression in 21 CRPC patients in order to study enza resistance mechanisms including lineage plasticity. 18 of 21 patients had the same tissue type biopsied at progression, and 8 patients had the exact same lesion biopsied twice. In 3 of 21 patients, transcriptomic analysis showed evidence of lineage plasticity. The authors identified pathways (including E2F1 and MYC) activated at baseline in these 3 “converter” patients compared to non-converters. They identified a 14-gene signature activated at baseline in converters vs. non-converters, which they hypothesized to be associated with risk of enza-induced lineage plasticity. They show the signature is associated with poor survival in two independent mCRPC cohorts, and that it is enriched in a previously published PDX model that underwent castration-induced lineage plasticity, in contrast to 9 other PDX models that did not. Finally they identified gene expression changes between baseline and progression in the three converters, including marked decrease in AR, HOXB13, and AR target genes. An increase at progression in immune related pathways including IFN γ , IFN α , and TNF α signaling was noted in both converter and non-converter patients.

Overall, this is an interesting study of a unique cohort of mCRPC patients for which matched metastatic biopsies before and after enzalutamide resistance are available for analysis. Although the patient numbers are small (only 3 “converter” patients), this is indeed a unique cohort, and the focus on determinants of enzalutamide-induced lineage plasticity towards a neuroendocrine phenotype is timely and would be of interest to the field. Although many of the findings are confirmatory of prior published results (e.g. importance of E2F1 and MYC gain and TP53 loss for lineage plasticity), the 14-gene lineage plasticity risk signature is novel and of potential interest and utility to the field. I do have several concerns that should be addressed, as follows:

- Line 198 and Suppl Table S4: Please provide additional details with regard to the 14-gene lineage plasticity risk signature derived from genes upregulated in baseline converter vs. non-converter samples. Specifically, although the 14 genes of the signature are named in Suppl Table S4, there is no information provided on the log-fold change and p-values for these genes from the differential gene expression analysis.

- Lines 255-259, Fig. 4C, Suppl. Fig. 4C: What genes/pathways, if any, were uniquely activated in converters? The text describe pathways that were activated in both converters and non-converters,

as well as pathways uniquely downregulated in non-converters, but does not describe those that were uniquely activated in converters.

- There is a lack of validation of the genes upregulated at progression in the converters. Fig. 4D provides validation at the protein level of several key genes downregulated at progression in the converters, but it is missing an example of any upregulated genes. It is not clear why INSM1 was selected as the protein marker for lineage plasticity to be assessed given that the RNA-seq data did not show any increased expression of this gene at progression in the converters. The IF should be repeated using a gene that was actually identified to be upregulated in converters at progression (e.g. any of the upregulated genes highlighted in the volcano plot in Fig. 4a).

- Table 1: It is interesting that one of the converters (subject 135) exhibited AR amplification at both baseline and progression and yet still underwent lineage plasticity accompanied by a decrease in AR expression and AR score. Please comment in the Discussion on this paradoxical finding.

- With regard to the 18 patients that did not undergo conversion, what are the postulated mechanisms of resistance? Many of these patients seemed to have either stable or increased AR activity with progression based on the VIPER and ARG10 analyses. Were there AR splice variants identified in these samples? Were there AR amplification or mutations? Were other pathways involved? Suppl. Fig. 4 shows some pathway analysis, but it would add tremendously to the paper if further analyses were performed.

- At several places (e.g. line 149, 157, etc.) the text refers to "WCDT cluster", while the figures (e.g. Fig. 2) refer to "Aggarwal cluster". Please use consistent naming of this cluster throughout the manuscript to avoid confusion.

- Figure 4B: Please describe in the figure legend the numbers and p-value listed to the right of the heat map.

- Figure 4D: Please provide additional details in the Methods regarding how the image analysis and signal intensity quantification was performed.

- Line 238: It seems the correct figure reference is Suppl Fig. 1C.

Reviewer #3 (Remarks to the Author): expertise in RNA-seq bioinformatics analysis

This elegant study by Westbrook et al., investigates the transcriptome of matched specimens from pre and post- enzalutamide treated castrate resistant prostate cancer (CRPC) patients to infer information on the mechanisms of prostate cancer (PC) progression and resistance to enzalutamide.

They show that the transcriptional profile of matched tumor specimens does not change dramatically when subjected to treatment but three tumors out of 21 underwent a process defined as "conversion" that the authors suggest to be the result of increased cancer cell plasticity and transdifferentiation. The authors generalise that this process occurs in about 14-17 % of PCs when

exposed to enza or castration . The Authors further derive a gene signature associated with transcriptional (and cell) plasticity, that can stratify patients exposed to AR signalling inhibitors for treatment outcome.

The hypothesis is exciting and analysis of the data is original . The initial results are incremental in respect to previous publications by the group (REFs 25,29) but the development of a lineage plasticity-associated signature is novel, noteworthy, and useful to characterise the biology of a subset of PCs.

Also, this study brings forward an important concept, which is the contribution of lineage plasticity in prostate cancer progression and drug resistance. However the way lineage plasticity is defined could be improved. The authors should emphasise that they infer the tumor “conversion” based on information of their transcriptome configuration. They use the Zhang et al., and the Kim et al., signatures that are presented as associated with plasticity. Yet these signatures define some transdifferentiation to basal-like cell types and AR-repressed genes, respectively, highlighting that the concept of plasticity here is heavily intertwined with transdifferentiation and, as shown in Suppl Fig 1C, to specific transdifferentiation toward a NEPC-like phenotype.

Further, the authors uncover that the transdifferentiation toward the NEPC-like phenotype leads to somewhat different “ending points” depending from the "transcriptional starting point (as shown by the fact that the three matched samples of interest did not cluster together -line 184). The author should consider discussing about transcriptional plasticity (or transcriptional conversion) - mediated cancer cell transdifferentiation rather cancer cell plasticity. This would mean expanding on discussing "susceptibility to transition (transcriptional) states" (lines 205 / 328-331).

Using PDXs the authors describe how one in particular, LTL331 undergoes such transcriptional transdifferentiation. When the mice bearing LTL331 are subjected to castration, the castration resistant derivative has high score for the 14 gene signature, although the castration resistant derivative does not present striking genetic alterations compared to the parental. Also, two of the three pre-post enza patient specimens from the SUC2 cohort presented fairly consistent genetics (lines 278-285). This suggests that the phenomenon observed here is not only at transcriptional reprogramming but also epigenetics and chromatin reprogramming (discussed also in lines 328-331). Divergent chromatin and transcriptional reprogramming upon resistance to enzalutamide was recently shown at single cell level in PMID: 34489465 and the authors should consider referencing this work.

The study is worthy of publication and here I have some suggestions for further analysis and deeper characterization of the observations

1- Unsupervised clustering with all samples should be shown like in Suppl Fig. 1 G for 1000 most varying genes. Do the sample cluster for biopsy site more than with only 500 genes considered in the analysis

2- E2F1 is the top differentially activated transcription factor in converters. Have the authors observed any changes in RB signalling in the converters? See PMID: 33879449.

3- Figure 3A and figure 4C: Along with activation of MYC targets (Fig 3A), MYC and its complex companion MAX are also among the top activated transcription factors (Fig 3B) in converters, suggesting a role of MYC in the conversion process. However in figure 4C MYC targets are down upon progression of the converters, which blurs the suggestion above. MYC was shown to antagonise AR activity (see PMID: 28412251). Were the AR-related scores in Supp Fig A, E, and F overlapping with AR-MYC antagonised genes according to PMID: 28412251? Also, to clarify the role of MYC in this conversation, would be useful to stain for c-MYC / n-MYC as in figure 4D? The patient 135 has a MYC amplification, does this confound the transcriptional analysis?

4- Fig 3F: The score for the 14 gene signature should be shown for all the PDXs before and after castration-induced lineage plasticity in a heat map

5- High risk primary hormone sensitive PC patients tend to be treated with ARSI now. I understand that a validation cohort for the 14-gene signature does not exist, but did the authors try to stratify the TCGA PRAD with the 14 gene signature? To establish the threshold for the risk score, the PRAD dataset could be displayed and compared to the other cohorts in Fig 3C, and the PRAD dichotomised accordingly. Alternatively a the highest percentile could be used. Wouldn't this analysis give an idea of the amount of patients in which transcriptional transdifferentiation could possibly be triggered if using ARSIs (discussed in lines 338-341 / lines 381-383)? Do the stratification is improved considering Gleason groups (GS 6, 7 and above 8)?

Other minor suggestions

Discussion lines 374-402 reads a bit redundant.

RESPONSE TO REVIEWER COMMENTS

Reviewer #1 (Remarks to the Author): expert in prostate cancer

In this well-written manuscript by Westbrook et al, the authors address an important question: do we see evidence of lineage plasticity via expression analysis in prostate cancers that are progressing on the next-generation AR inhibitor enzalutamide? The authors are to be commended for performing bulk RNA sequencing on microdissected tumors at baseline and at progression from 21 patients (same tumor biopsied for 8/21 patients) – this is no easy task and is the largest dataset of its kind. One of the most interesting findings is a negative one: most tumors (18/21) did not show a substantial change of transcriptional profile at progression, and half maintained high AR profile which is consistent with clinical observations: most patients with prostate cancer still die of high PSA producing disease. Nonetheless, lineage plasticity remains important biologically and clinically given its associated aggressive clinical behavior. 3/21 patients, called ‘converters’ had evidence of transformation to a lineage plasticity phenotype based on change in their expression profiles at progression. The authors characterize transcriptional changes for those 3 patients and derive an exploratory signature from their baseline tumors that may be associated with risk of developing a lineage plasticity phenotype. I believe this is important novel work that merits publication with some minor comments:

Rev 1, Comment 1: In supplementary data, the authors should provide patient level clinical and genomic information in table format (ie sample type, enzalutamide response characteristics, prior treatment and most importantly RNAseq results and DNA sequencing when available). The summary tables do not provide sufficient detail.

Response: Thank you for this comment. Table S2 has patient level clinical data that include biopsy site, time on enzalutamide and prior treatments. A new Supplemental Table 9, has been added that included mutation calls for each sample. A supplemental data file has been added that has TPM expression values for each sample. Finally, we are depositing raw sequencing data to the European Genome-phenome Archive (EGA).

Rev 1, Comment 2: As the authors note in the discussion, overall enzalutamide response in this 21 patient cohort is generally poor despite relatively small proportion of patients receiving prior abiraterone or taxane therapy. Suggest commenting on this in the results section too.

Response: Thank you for this suggestion. As the reviewer notes, we discuss possible reasons for the poorer outcome in our cohort in the text. These patients included in our analysis had lesions that were amenable to biopsy at baseline and progression, and many were symptomatic. Thus, they may have had more widespread disease than those enrolled on the PREVAIL clinical trial of enzalutamide who were largely asymptomatic. Unlike PREVAIL, our cohort also included several subjects with prior exposure to abiraterone or docetaxel. It is probable that a combination of these factors contributed to the PFS and response rates observed. We have added a comment in the results section page 5, line 130 as recommended.

Rev 1, Comment 3: In the Fig 1 clustering analysis, were the ‘closest’ match pairs the cases where the same site was biopsied? Did tumors cluster together based on site of metastasis? Curious to know if microdissection mostly removes transcriptional effect of microenvironment.

Response: Thank you for this important comment. At your suggestion, we examined this question. Biopsying the same lesion was not significantly associated with being nearest neighbor: 5/13 cases that were nearest neighbors had the same lesion biopsied vs. 3/8 cases that were not nearest neighbors and had the same lesion biopsied ($p=0.30$). We have updated the text to reflect this (excerpted below- please see page 5, lines 134-137). As seen in Figure 2A samples do not cluster together based solely on metastatic site. Together, this suggest that much of the microenvironment has been removed with LCM. The text was updated to reflect this point.

“Samples did not cluster together based solely on the site of biopsy, indicating laser capture microdissection removed much of the microenvironment from these samples. Furthermore, whether the same lesion was biopsied did not impact how samples clustered.”

Rev 1, Comment 4: Lines 178-179: authors suggest that the conversion rate seen here (3/21 = 14%) nearly matches the frequency of 17% reported in Aggarwal et al JCO. I do not think this is a good comparison: Aggarwal et al focused on IHC to define treatment emergent NEPC and found a surprisingly high rate of 17%. As the authors state here, the 3 converters did not show transcriptional evidence of NEPC and no samples displayed significant NEPC features at progression despite the aggressive behavior of this cohort. The authors should modify this statement and comment on the absence of NEPC features in their cohort.

Response: Thank you for this comment. Of note, Aggarwal, et al used histology—rather than IHC—to define NEPC. We used RNA-seq in the Aggarwal paper to define clusters and determined that cluster 2 was enriched for AR activity low tumors that include NEPC and double negative prostate cancer. Cluster 2 comprised 10% of cases in that report. As we point out in this manuscript, our converter tumors also harbored a cluster 2 program, and the frequency of 3/21 closely matches the frequency of cluster 2 tumors in the Aggarwal report. We have amended the text on page 7, lines 185-186 to focus on cluster 2, rather than NEPC. The text is excerpted below:

“Altogether, these results suggest that enza-induced lineage plasticity and conversion to an AR-independent program occurs in a subset of tumors (3/21 or 14%), similar to the frequency of cluster 2 tumors (10%) described by Aggarwal previously²⁵.”

Rev 1, Comment 5: The authors derive a 14 gene expression signature from the (only) 3 converters which they suggest is associated with increased risk of lineage plasticity. They show that this signature is associated with worse outcomes in independent cohorts, and that it is associated with lineage plasticity in a PDX model. Evidently this is very exploratory, but to try to address the potential future utility of such a signature, can the authors comment in the discussion on how such a signature may be used in the context of other molecular profiles e.g. TP53/RB1/PTEN status, and how a marker of lineage plasticity risk may be clinically useful?

Rev 1, comment 6: Along the same line, it is surprising to see that none of the converters have compound tumor suppressor genomic alterations (Table 1) that so far are considered to be the strongest markers of developing lineage plasticity. Can the authors comments on mutational status in TP53/RB1/PTEN in the other tumors in their cohort, as well as on mutational status in the signature positive cases in the independent cohorts (Fig3D-E)?

Response: These are important questions. It is clear that loss of TP53/RB1/PTEN are common in NEPC and that their loss accelerates NEPC lineage plasticity in experimental mouse models. The importance of the loss of these genes for lineage switching to DNPC remains to be determined.

We used a targeted DNA sequencing panel that may miss namely structural variants that are known to disable the function of tumor suppressor genes, including TP53/RB1/PTEN. A previous WCDT publication (PMID 30033370) performed whole genome sequencing and several of our converter patients' baseline tumors underwent whole genome sequencing (80 and 210). Patient 80's baseline tumor had a TP53 structural variant while patient 210's baseline tumor did not have structural variants in TP53/RB1/PTEN. Patient 135's baseline tumor did not undergo WGS. These data suggest that structural variants in these tumor suppressors may have contributed to risk of lineage plasticity, at least in subject 80.

Importantly, in response to concern #6 below, we examined TP53/RB1/PTEN alterations in patients with high lineage plasticity risk scores in the Abida and Alumkal datasets that were used as independent cohorts in our paper. Two patients out of six in the Abida dataset with high converter scores had loss of 2/3 of TP53/RB1/PTEN; only one patient out of three with high converter risk in the Alumkal dataset had available DNA testing and showed no mutations. These data are displayed in Supplemental Table 8. These data strongly suggest that TP53/RB1/PTEN loss are not the main drivers of activation of the lineage plasticity risk signature we identified. We have added text to the results page 12, lines 311-321 and discussion page 15 lines 385-395 regarding these points and excerpted below.

Regarding the lineage plasticity risk signature, appropriate validation cohorts do not exist. However, we are currently attempting to build them. In the future, it will be important to determine whether combining our gene expression signature and TP53/RB1/PTEN will be better than either alone for identifying tumors at risk of lineage plasticity. We have added points in the discussion about the potential for examining the lineage plasticity risk signature and TP53/RB1/PTEN status. Please see page 16, lines 398-400. Text is excerpted below.

In regard to mutations in the wider cohort, we have added DNA alterations for this matched biopsy cohort as a whole in Supplemental Table 9. Among non-converters, one patient that did not convert had compound loss of TP53/RB1 at baseline and progression and another 2 patients had compound loss of TP53/PTEN at progression.

From Results

"Loss of the tumor suppressor genes TP53, RB1, and PTEN has been linked to lineage plasticity risk in pre-clinical models^{32,33}. However, we do not know if the presence of these genomic abnormalities in patient tumors is associated with risk of lineage plasticity to DNPC. One of the three converter patients (subject 80) was found to have an inactivating PTEN mutation and a second patient (subject 103) had RB1 loss, but none were found to have compound TP53/RB1/PTEN loss. When available, we also examined TP53/RB1/PTEN status for tumors from the Abida, et al.¹⁰ and Alumkal, et al.¹⁸ cohorts that had high lineage plasticity risk scores. Of the seven high lineage plasticity risk score tumors examined from these two validation cohorts, only two tumors had loss of two or more of the genes TP53, RB1, and PTEN (Supplemental Table 8). DNA-sequencing of matched metastatic biopsies for the cohort as a whole is shown in Supplemental Table 9."

From Discussion

“Loss of the tumor suppressor genes TP53, RB1, and PTEN has been linked to NEPC lineage plasticity risk in pre-clinical models^{32,33}. Importantly, our three converter tumors underwent lineage switching to DNPC, rather than NEPC. TP53 was inferred to be the most deactivated master regulator in the baseline biopsies from the converters, and we determined that an RB1 loss signature was increased at progression in converters. Two out of four of the tumors from the converter patients in our matched biopsy cohort harbored genetic alterations in at least one of these genes by the assays used in this paired analysis. These genes can also be lost via structural alterations that we did not measure in our DNA sequencing assay or via non-genetic mechanisms^{31,46}. In fact, we previously performed whole genome sequencing on the baseline tumor from one of the converters (subject 80) and found a structural alteration in TP53 using that approach⁴⁷. However, in examining the TP53, RB1, and PTEN status for tumors from the Abida, et al.¹⁰ and Alumkal, et al.¹⁸ cohorts, we determined that only two of the seven tumors with high lineage plasticity risk scores had loss of two or more of these genes. In the future, it will be important to determine whether combining TP53, RB1, and PTEN status with the gene signature we identified is better than either alone for identifying tumors at risk of lineage plasticity.”

Reviewer #2 (Remarks to the Author): expert in prostate cancer biomarkers

This is an interesting manuscript in which the authors performed laser capture microdissection and RNA-seq on matched metastatic tumor biopsies prior to enzalutamide and at the time of progression in 21 CRPC patients in order to study enza resistance mechanisms including lineage plasticity. 18 of 21 patients had the same tissue type biopsied at progression, and 8 patients had the exact same lesion biopsied twice. In 3 of 21 patients, transcriptomic analysis showed evidence of lineage plasticity. The authors identified pathways (including E2F1 and MYC) activated at baseline in these 3 “converter” patients compared to non-converters. They identified a 14-gene signature activated at baseline in converters vs. non-converters, which they hypothesized to be associated with risk of enza-induced lineage plasticity. They show the signature is associated with poor survival in two independent mCRPC cohorts, and that it is enriched in a previously published PDX model that underwent castration-induced lineage plasticity, in contrast to 9 other PDX models that did not. Finally they identified gene expression changes between baseline and progression in the three converters, including marked decrease in AR, HOXB13, and AR target genes. An increase at progression in immune related pathways including IFN γ , IFN α , and TNF α signaling was noted in both converter and non-converter patients.

Overall, this is an interesting study of a unique cohort of mCRPC patients for which matched metastatic biopsies before and after enzalutamide resistance are available for analysis. Although the patient numbers are small (only 3 “converter” patients), this is indeed a unique cohort, and the focus on determinants of enzalutamide-induced lineage plasticity towards a neuroendocrine phenotype is timely and would be of interest to the field. Although many of the findings are confirmatory of prior published results (e.g. importance of E2F1 and MYC gain and TP53 loss for lineage plasticity), the 14-gene lineage plasticity risk signature is novel and of potential interest and utility to the field. I do have several concerns that should be addressed, as follows:

Rev 2, comment 1: Line 198 and Suppl Table S4: Please provide additional details with regard to the 14-gene lineage plasticity risk signature derived from genes upregulated in baseline converter vs. non-converter samples. Specifically, although the 14 genes of the signature are named in Suppl Table S4, there is no information provided on the log-fold change and p-values for these genes from the differential gene expression analysis.

Response: Thank you for your constructive comment. We have included the fold-change and p-values for the genes that comprise this signature in a revised Supplemental Table 4.

Rev 2, comment 2: Lines 255-259, Fig. 4C, Suppl. Fig. 4C: What genes/pathways, if any, were uniquely activated in converters? The text describe pathawys that were activated in both converters and non-converters, as well as pathways uniquely downregulated in non-converters, but does not describe those that were uniquely activated in converters.

Response: Thank you for this comment. We have amended the text to more clearly state the pathways that are uniquely activated in the converters vs. the non-converters at progression. The unique pathways activated in the converters at progression include: allograft rejection, IL6-JAK-STAT3 signaling, inflammatory response and complement. All of these are consistent with a more stem-like program in the progression samples from converters vs. non-converters that appear more AR-driven. Please see page 11, lines 280-284. Text is excerpted below.

“Uniquely downregulated pathways in the progression samples from non-converters included: E2F targets, G2M checkpoint, and hedgehog signaling. The only uniquely upregulated pathway in non-converters was protein secretion while uniquely downregulated pathways included hedgehog signaling, G2M checkpoint and E2F targets.”

Rev 2, comment 3: There is a lack of validation of the genes upregulated at progression in the converters. Fig. 4D provides validation at the protein level of several key genes downregulated at progression in the converters, but it is missing an example of any upregulated genes. It is not clear why INSM1 was selected as the protein marker for lineage plasticity to be assessed given that the RNA-seq data did not show any increased expression of this gene at progression in the converters. The IF should be repeated using a gene that was actually identified to be upregulated in coverters at progression (e.g. any of the upregulated genes highlighted in the volcano plot in Fig. 4a).

Response: Thank you for this important comment regarding selection of markers used in the multiplex immunofluorescence (IF) assays. When we applied gene signatures developed by Labrecque, et al to our matched RNA-seq data, all baseline samples were classified as AR-driven prostate cancer (ARPC). However, the progression samples from the three converters were classified as double negative prostate cancer (DNPC) and not NEPC. In keeping with the DNPC classification, we did not find that transcripts linked to NEPC were highly-upregulated in the progression samples from converters (Supplemental Table 6). The main intent of our multiplex IF assays was to confirm loss of the ARPC phenotype as exemplified by loss of AR, NKX3.1, and HOXB1 and that these tumors did not have NEPC marker expression. To rule out the latter, we measured the NEPC marker INSM1. The lack of AR expression and absence of INSM1 are pathognomonic for DNPC. Importantly, our IF results confirm our RNA-seq results and strongly suggest the converters have a DNPC phenotype. Testing for a gene upregulated in the converters (there are several good candidates in Supplemental Table 6) would provide protein confirmation of the RNA-seq

data. Regrettably, we have exhausted all the blocks for these cases, and so IF for other markers is not possible.

Rev 2, comment 4: Table 1: It is interesting that one of the converters (subject 135) exhibited AR amplification at both baseline and progression and yet still underwent lineage plasticity accompanied by a decrease in AR expression and AR score. Please comment in the Discussion on this paradoxical finding.

Response: Thank you for this comment. We agree that it is interesting that subject 135 had AR amplification in both the progression and baseline sample. These data strongly suggest that the progression sample underwent lineage plasticity from a baseline tumor population that was once AR-driven. However, the AR expression clearly declined in his progression tumor (Figure 2B). There are several possible explanations for the reduced AR mRNA expression at progression, including repressive histone methylation or DNA methylation. We have amended the discussion to highlight the importance of the finding of AR amplification and mechanisms that might explain AR expression loss. Please see page 16, lines 403-408. Text is excerpted below.

“It is striking that the baseline and progression sample from one patient (subject 135) whose tumor underwent conversion had AR amplification despite loss of AR expression at progression. These data strongly suggest that the progression sample underwent lineage plasticity from a baseline tumor population that was once AR-driven. There are several possible explanations for loss of AR expression despite the AR amplification, including repressive histone methylation or DNA methylation^{32,48}.”

Rev 2, comment 5: With regard to the 18 patients that did not undergo conversion, what are the postulated mechanisms of resistance? Many of these patients seemed to have either stable or increased AR activity with progression based on the VIPER and ARG10 analyses. Were there AR splice variants identified in these samples? Were there AR amplification or mutations? Were other pathways involved? Suppl. Fig. 4 shows some pathway analysis, but it would add tremendously to the paper if further analyses were performed.

Response: We thank the reviewer for this insightful comment. Mechanisms of resistance in our cohort are heterogeneous, but many tumors remain reliant on AR signaling at progression as others studying ARSI resistance have found (PMID 33664492, 33849963).

As the reviewer mentions, three non-converter patients had increased AR signaling, 9 non-converter patients had similar and 6 non-converter patients had decreased AR signaling at progression by VIPER score. Two patients had AR amplifications detected at progression that were not detected at baseline (new Supplemental Table 8).

AR-V7 expression is shown below. There is clear decrease in ARV7 in the converter samples and mixed findings in the overall cohort without a significant trend overall ($p=0.56$). We have included this in results page 6, lines 144-146 (excerpted below) and as a new Supplemental Figure 2C (shown below).

“Though AR-V7 expression increased in several samples at progression, the difference in expression using the entire 21-patient cohort was not statistically significant (Supplemental Figure 1C).”

Hallmark pathway analysis is shown in Supplemental Figure 4. Importantly, while there are shared pathways activated in progression samples from converters and non-converters, the only unique pathway activated solely in non-converters was protein secretion while hedgehog signaling, G2M checkpoint and E2F target pathways were downregulated. We have amended the discussion to specifically call out this unique pathway. See excerpted text from page 11, line 280-284 shown below:

“Uniquely downregulated pathways in the progression samples from non-converters included: E2F targets, G2M checkpoint, and hedgehog signaling. The only uniquely upregulated pathway in non-converters was protein secretion while uniquely downregulated pathways included hedgehog signaling, G2M checkpoint and E2F targets.”

Rev 2, comment 6: At several places (e.g. line 149, 157, etc.) the text refers to “WCDT cluster”, while the figures (e.g. Fig. 2) refer to “Aggarwal cluster”. Please use consistent naming of this cluster throughout the manuscript to avoid confusion.

Response: Thank you for this comment and the opportunity to clarify the cluster designation for readers. We have changed all instances of “WCDT” cluster to “Aggarwal” cluster.

Rev 2, comment 7: Figure 4B: Please describe in the figure legend the numbers and p-value listed to the right of the heat map.

Response: Thank you for this comment. The figure legend has been updated to the following:

“ARG10 gene signature heatmap for three converters at baseline and progression. The left half shows the expression levels of individual genes in the ARG10 signature, and the right half shows the ARG10

signature score. p-value shown is for a paired t-test between baseline and progression ARG10 scores (n=3 pairs).”

Rev 2, comment 8: Figure 4D: Please provide additional details in the Methods regarding how the image analysis and signal intensity quantification was performed.

Response: Thank you for this comment, we have added additional methods regarding the image analysis as suggested to the methods. Please see page 21 , lines 528-532. Text is excerpted below.

“For digital image analyses, fluorescence images were loaded into QuPath (v0.3.0) 56. Regions of interest containing cancer cells were outlined using the polygon annotation tool, and the positive cell detection feature was used to segment individual cells in all channels. The mean nuclear intensity was extracted for each single cell using the detection measurements function for each channel separately.”

Rev 2, comment 9: Line 238: It seems the correct figure reference is Suppl Fig. 1C.

Response: We apologize for this error. Thank you for catching it. We have corrected it.

Reviewer #3 (Remarks to the Author): expertise in RNA-seq bioinformatics analysis

This elegant study by Westbrook et al., investigates the transcriptome of matched specimens from pre and post- enzalutamide treated castrate resistant prostate cancer (CRPC) patients to infer information on the mechanisms of prostate cancer (PC) progression and resistance to enzalutamide.

They show that the transcriptional profile of matched tumor specimens does not change dramatically when subjected to treatment but three tumors out of 21 underwent a process defined as “conversion” that the authors suggest to be the result of increased cancer cell plasticity and transdifferentiation. The authors generalise that this process occurs in about 14-17 % of PCs when exposed to enza or castration . The Authors further derive a gene signature associated with transcriptional (and cell) plasticity, that can stratify patients exposed to AR signalling inhibitors for treatment outcome.

The hypothesis is exciting and analysis of the data is original . The initial results are incremental in respect to previous publications by the group (REFs 25,29) but the development of a lineage plasticity-associated signature is novel, noteworthy, and useful to characterise the biology of a subset of PCs.

Also, this study brings forward an important concept, which is the contribution of lineage plasticity in prostate cancer progression and drug resistance. However the way lineage plasticity is defined could be improved. The authors should emphasise that they infer the tumor “conversion” based on information of their transcriptome configuration. They use the Zhang et al., and the Kim et al., signatures that are presented as associated with plasticity. Yet these signatures define some transdifferentiation to basal-like cell types and AR-repressed genes, respectively, highlighting that the concept of plasticity here is heavily intertwined with transdifferentiation and, as shown in Suppl Fig 1C, to specific transdifferentiation toward a NEPC-like phenotype.

Rev 3, comment 1: Further, the authors uncover that the transdifferentiation toward the NEPC-like phenotype leads to somewhat different “ending points” depending from the "transcriptional starting point (as shown by the fact that the three matched samples of interest did not cluster together -line 184). The author should consider discussing about transcriptional plasticity (or transcriptional conversion) - mediated cancer cell transdifferentiation rather cancer cell plasticity. This would mean expanding on discussing "susceptibility to transition (transcriptional) states" (lines 205 / 328-331).

Response: Thank you for this thought-provoking comment. We use the term lineage plasticity as defined by the NCI Workshop on Lineage Plasticity and Androgen Receptor-Independent Prostate Cancer group consensus (PMID 31363002). While another commonly cited definition (PMID 32152485) proposes lineage plasticity as the ability to transition from one committed developmental pathway to another, the NCI Workshop consensus definition includes the transition process in similar terms. We have clarified in the specific areas suggested but continue to use lineage plasticity broadly throughout the rest of the work. Please see page 9, lines 214-216 and page 14-15 lines 366-370. The latter text is excerpted below.

“Importantly, we found that DNA mutations and copy number alterations were conserved between samples from these converters. These results suggest that the marked differences in gene expression observed and transcriptional conversion in these tumors may be epigenetically regulated, rather than due to selection of an independent, genetically distinct clone that is different from the baseline bulk tumor population.”

Rev 3, comment 2: Using PDXs the authors describe how one in particular, LTL331 undergoes such transcriptional transdifferentiation. When the mice bearing LTL331 are subjected to castration, the castration resistant derivative has high score for the 14 gene signature, although the castration resistant derivative does not present striking genetic alterations compared to the parental. Also, two of the three pre-post enza patient specimens from the SUC2 cohort presented fairly consistent genetics (lines 278-285). This suggests that the phenomenon observed here is not only at transcriptional reprogramming but also epigenetics and chromatin reprogramming (discussed also in lines 328-331). Divergent chromatin and transcriptional reprogramming upon resistance to enzalutamide was recently shown at single cell level in PMID: 34489465 and the authors should consider referencing this work.

Response: Thank you for this suggestion. As stated, this suggested reference provides insight into epigenetic changes which may underlie transcriptional reprogramming when model cell lines become resistant to enzalutamide. This is highly relevant to this work, as an analogous process may contribute to the transcriptional transdifferentiation/lineage plasticity process we describe in converter patients. We have added the citation to the discussion section as suggested. Please see page 14, line 363-365. Text excerpted below:

“Recent work by Taavitsainen, et al. and Han, et al. demonstrates that enzalutamide induces transcriptional reprogramming of prostate cancer models—an effect that is partially mediated by changes in chromatin structure^{45,46}.”

Rev 3, comment 3: The study is worthy of publication and here I have some suggestions for further

analysis and deeper characterization of the observations. Unsupervised clustering with all samples should be shown like in Suppl Fig. 1 G for 1000 most varying genes. Do the sample cluster for biopsy site more than with only 500 genes considered in the analysis

Response: Thank you for this comment. We performed unsupervised clustering for the 1000 most varying genes. Results are now included in a new Supplemental Figure 1H-I (below). Using 1,000 genes, rather than 500 genes, did not lead to clustering of samples together based on biopsy site.

Rev 3, comment 4: E2F1 is the top differentially activated transcription factor in converters. Have the authors observed any changes in RB signalling in the converters? See PMID: 33879449.

Response: Thank you for this important question. To address this concern, we applied a well-studied RB1 loss (PMID: 31010837) to our entire cohort in a new Supplemental Figure 1J (below). We find that the RB1 loss signature is not more highly activated in the baseline tumors from converters vs. non-converters. However, there was a trend to increase in this RB1 loss signature in the progression samples from the converters ($p=0.089$). We did not detect new RB1 mutations at progression in the converters, suggesting that RB1 loss may be occurring through other mechanisms. We have added this point to the text on page 8, lines 200-203 and page 15, line 387-389. Text is excerpted below:

“Additionally, we found that there was an upward trend in a previously described RB1 loss signature³¹ in the progression samples from converters, further suggesting E2F1 activation contributes to the lineage switch (Supplemental Figure 1J).”

“TP53 was inferred to be the most deactivated master regulator in the baseline biopsies from the converters, and we determined that an RB1 loss signature was increased at progression in converters.”

Rev 3, comment 5: Figure 3A and figure 4C: Along with activation of MYC targets (Fig 3A), MYC and its complex companion MAX are also among the top activated transcription factors (Fig 3B) in converters, suggesting a role of MYC in the conversion process. However in figure 4C MYC targets are down upon progression of the converters, which blues the suggestion above. MYC was shown to antagonise AR activity (see PMID: 28412251). Were the AR-related scores in Supp Fig A, E, and F overlapping with AR-MYC antagonised genes according to PMID: 28412251? Also, to clarify the role of MYC in this conversation, would be useful to stain for c-MYC / n-MYC as in figure 4D? The patient 135 has a MYC amplification, does this confound the transcriptional analysis?

Response: Thank you for this comment. We are aware of the work cited by the reviewer by the Mills Lab. AR interactions with MYC are clearly complex. Of note, we previously determined that AR promotes c-Myc expression in a ligand-independent manner (PMID: 23704919). Thus, one possible explanation for c-Myc pathway loss at progression is due to loss of AR function.

We have performed the analyses requested examining a 365 gene signature of Myc-antagonized genes (shown below). The differences in these signature scores between progression and baseline were non-significant for the cohort as a whole and for the converters. This is in contradistinction to the statistically significant reduced AR activity measured by ARG10, VIPER, and the Kim, et al AR-repressed signatures that are shown in Supplemental Figure 1A, E and F.

Regrettably, we have exhausted all the blocks for these cases, and so IHC for Myc family members is not possible.

Regarding the amplification found in subject 135, this amplification in this subject does not appear to be the driving factor for our analysis implicating activation of Myc in the baseline samples from the converters vs. non-converters. Importantly, we examined Myc mRNA expression in our cohort, and all three baseline samples from converters had high Myc expression. Importantly, Myc expression declined in all three converters (see below plots that are included in this response document only and not the manuscript). Thus, the loss of AR activity we identified in the three converters occurs concomitantly with loss of Myc expression and supports our prior work demonstrating that AR activation is linked to Myc expression.

Rev 3, comment 6: Fig 3F: The score for the 14 gene signature should be shown for all the PDXs before and after castration-induced lineage plasticity in a heat map

Response: Thank you for this comment. Gene expression profiling data is available for all the baseline hormone-naïve PDXs but just one CRPC PDX from this PDX cohort—the castration-derivative of LTL331 called LTL331R that undergoes lineage switching. We have examined the 14 gene signature in all adenocarcinoma samples. Importantly, LTL331 had the highest signature score. We have added a dot plot with LTL331 and all other hormone naïve LTL adenocarcinoma PDXs with the high vs. low-risk score cutoff from our matched biopsy patients to the main figure (Figure 3F, below). Only LTL331 had a high score.

A heatmap for all the PDXs, including the LTL331R PDX that undergoes lineage switching, is shown in new Supplemental Figure 2C (see below). We have moved the GSEA plot we showed originally to Supplemental Figure 2D.

Rev 3, comment 7: High risk primary hormone sensitive PC patients tend to be treated with ARSI now. I understand that a validation cohort for the 14-gene signature does not exist, but did the authors try to stratify the TCGA PRAD with the 14 gene signature? To establish the threshold for the risk score, the PRAD dataset could be displayed and compared to the other cohorts in Fig 3C, and the PRAD dichotomised accordingly. Alternatively a the highest percentile could be used. Wouldn't this analysis give an idea of the amount of patients in which transcriptional transdifferentiation could possibility be triggered if using ARSIs (discussed in lines 338-341 / lines 381-383)? Do the stratification is improved considering Gleason groups (GS 6, 7 and above 8)?

Response: Thank you for this suggestion. We also examined the 14 gene signature in the TCGA and found that only 2 samples (one Gleason 7 and another Gleason 9) out of 495 have a high conversion risk score (new Supplemental Figure 2B). This suggests that this signature is not significantly enriched in the vast majority of primary tumors and that this signature may not be suitable for identifying lineage plasticity risk in primary tumors. We have included a dot plot of the TCGA data in a new Supplemental Figure 2B (see below) and excerpted text from page 9, lines 230-234 below.

“To determine if the lineage plasticity risk signature was activated in primary tumors, we examined the TCGA dataset³⁹. Importantly, only two of 495 patients had high risk scores (Supplemental Figure 2B). The lower frequency in primary tumors vs. CRPC cohorts suggests that activation of this lineage plasticity risk program may be induced by castration.”

Progression-free survival analysis of these two high scoring tumors vs. the other 493 does not show a significant difference (see below). We have included the progression free survival data in this response document only and not the manuscript because of the small proportion of patients who have a high-risk score and the non-significant association with progression-free survival.

Rev 3, comment 8:

Other minor suggestions

Discussion lines 374-402 reads a bit redundant.

Response: This passage has been streamlined as suggested, thank you.

REVIEWERS' COMMENTS

Reviewer #1 (Remarks to the Author):

All of my comments were thoroughly addressed by the authors. This is an insightful manuscript.

Reviewer #2 (Remarks to the Author):

The authors are to be commended for responding to the reviewer comments in a thoughtful and thorough fashion. All of my concerns have been addressed satisfactorily.

Reviewer #3 (Remarks to the Author):

The authors have addressed my comments. This is an elegant study.

Well done to all.